# Neural variability reliably encodes interindividual differences in the perception of pain intensity

**Li-Bo Zhang**[1,2,3☯*], **Xin-Yi Geng**[1,2☯], **Li Hu**[1,2*]

**1** State Key Laboratory of Cognitive Science and Mental Health, Institute of Psychology, Chinese Academy of Sciences, Beijing, China, **2** Department of Psychology, University of Chinese Academy of Sciences, Beijing, China, **3** Neuroscience and Behaviour Laboratory, Italian Institute of Technology, Rome, Italy

☯ These authors contributed equally to this work.
* shelling.libo.zhang@gmail.com (LBZ); huli@psych.ac.cn (LH)

## Abstract

Neural activity varies dramatically across time. While such neural variability has been associated with cognition, its relationship with pain remains largely unexplored. Here, we systematically investigated the relationship between neural variability and pain, particularly individual differences in pain intensity discriminability, in six large electroencephalography (EEG) datasets (total $N=633$), where healthy volunteers (Datasets 1–5; $N=606$) and postherpetic neuralgia patients (Dataset 6; $N=27$) received painful or nonpainful sensory stimuli. We found robust correlations between neural variability and interindividual pain intensity discriminability. These correlations were replicable in multiple datasets and seemed not to be caused by stimulus-general factors, as no significant correlations were observed in nonpain modalities. Importantly, variability and amplitude of EEG responses were mutually independent and had distinct temporal and oscillatory profiles in encoding pain intensity discriminability. These findings demonstrate that neural variability is a replicable and potentially preferential indicator of individual differences in pain intensity discriminability, thereby enhancing the understanding of neural encoding of pain intensity discriminability and underscoring the value of neural variability in pain studies.

## Introduction

Almost all pain studies characterize nociceptive-evoked neural activity in terms of its amplitude. For example, the most common electroencephalography (EEG) metrics involve the amplitude of nociceptive-evoked potentials, such as the N2 and P2 components [1–5]. Although this approach has remarkably advanced our understanding of pain processing, it overlooks another critical aspect of neural activity: its variability. Importantly, neural variability has been increasingly recognized as having a functional role in some fields, such as sensory discrimination, perception, and memory [6–8]. A positive correlation has been found between increased variability in functional

**Data availability statement:** The data and code to replicate all results are available on the Open Science Framework (https://doi.org/10.17605/OSF.IO/QTV8A).

**Funding:** This work was supported by National Key Research and Development Program of China (2023YFC2508702 to L.H.), National Natural Science Foundation of China (32071061 to L.H.), and Beijing Natural Science Foundation (JQ22018 to L.H.; L246074 to L.H.). These funders played no role in the study design, data collection and analysis, decision to publish, or preparation of the manuscript.

**Competing interests:** The authors have declared that no competing interests exist.

**Abbreviations:** AUC, area under the curve; BF, Bayes factor; BOLD, blood-oxygenation-level-dependent; CV, coefficient of variation; EEG, electroencephalography; ERPs, event-related potentials; FDR, false discovery rate; fMRI, functional magnetic resonance imaging; HV, healthy volunteers; ICA, independent component analysis; JND, just-noticeable difference; NRS, numerical rating scale; PE, permutation entropy; PHN, postherpetic neuralgia; SD, standard deviation; SDT, signal detection theory; SPL, sound pressure level.

magnetic resonance imaging (fMRI) signals within the human motion complex and enhanced discrimination of visual motion direction [9]. In addition, neural variability in EEG signals has also been associated with interindividual differences in reaction time during face-recognition tasks [10]. Thus, neural variability may provide new insights into how the brain represents pain information beyond the traditional amplitude-based approach.

Previous amplitude-based research has greatly contributed to revealing the neural encoding of pain [11–14]. Pain variations at intraindividual and interindividual levels have been associated with the amplitudes of N2, P2, and gamma-band oscillations [3,15–18]. However, EEG responses often correlate with perceptual variation in nonpain sensory modalities as well [15,19]. Interestingly, interindividual pain intensity discriminability, the ability to distinguish the intensity between two or more painful stimuli, appears to be preferentially encoded by the amplitude of event-related potentials (ERPs). No reliable correlations have been found between individual differences in intensity discriminability and ERPs in nonpain modalities [20,21]. This potential preferential encoding is particularly noteworthy since pain intensity discriminability appears clinically relevant. Some small studies have observed impaired pain intensity discriminability in patients with chronic pain [22,23] and sickle cell disease [24], and better pain intensity discriminability predicted the effect of pain treatment [25,26]. Some studies have shown that tactile discrimination training could reduce cortical reorganization and pain symptoms in chronic pain patients [27–29], suggesting a potential link between discrimination ability and chronic pain. In reality, however, pain intensity discriminability remains relatively underexplored [15,16,21,30,31].

In contrast to the amplitude-based approach, few, if any, studies have systematically examined how neural variability contributes to the neural encoding of pain, particularly individual differences in pain intensity discriminability. Many fundamental questions remain unanswered. First, it is unclear whether neural variability reliably reflects pain. Currently, only a handful of studies have indirectly associated neural variability with pain, such as with temporal summation of pain [32,33]. The relationship between neural variability and pain, particularly pain intensity discriminability, thus remains largely unknown. The replicability of the association also needs to be tested, especially when the replication crisis has called into question even classical findings in biological sciences [34–37]. Second, it is unclear whether and how the role of neural variability in encoding pain intensity discriminability differs from that of the amplitude of neural activity. Given the association between ERP amplitude and pain intensity discriminability [21], this question is key to establishing the independent role of neural variability as an indicator of pain intensity discriminability, above and beyond the amplitude of neural activity. Third, it has yet to be determined whether the association between neural variability and pain intensity discrimination is caused by sensory stimulus-general factors. Stimulus-general confounders such as salience have been shown to contribute to the relationship between ERP amplitude and pain [19,38,39].

To address these questions, we systematically examined the relationship between neural variability and pain, with a particular focus on individual differences in pain

intensity discriminability, using six large EEG datasets (total $N = 633$; see Fig 1). Datasets 1–5 included healthy volunteers exposed to painful laser stimuli and nonpainful tactile, auditory, and visual stimuli of varying intensities, while Dataset 6 included chronic pain patients receiving painful laser stimuli. We focused on ERPs because pain-preferential gamma-band oscillations normally have lower signal-to-noise ratio than ERP components like N2 and P2 [40,41]. Moreover, different from ERP amplitude, event-related spectral perturbations such as α oscillations did not show reliable associations with pain intensity discriminability in previous studies [21]. Although our main focus was on individual differences in pain intensity discriminability, we also explored the relationship between neural variability and intraindividual pain intensity perception and interindividual pain intensity sensitivity.

## Results

### Neural variability reliably encodes pain intensity discriminability independent of amplitude

We mainly employed temporal standard deviation (SD) calculated with 100-ms sliding windows as the primary metric for neural variability in this study (Fig 1C) and verified the robustness of our findings with windows of other lengths. Here, individual differences in pain intensity discriminability were operationalized as differences in mean subjective ratings between low- and high-intensity stimuli. For example, if one person rates two painful stimuli as 3 and 5 on average on a 0~10 rating scale, their pain intensity discriminability is quantified as $5 - 3 = 2$. Essentially, this quantity is the slope of the psychophysical function given two fixed and medium physical intensities. Individuals who have better intensity discriminability will have a steeper psychophysical function and thus larger rating differences (S1 Fig). Traditionally, just-noticeable difference (JND) is used to measure sensory discrimination rather than rating differences. JND and rating differences are conceptually related. The former holds constant pain sensation and uses the difference in stimulus intensities to quantify the ability of individuals to discriminate between them. On the other hand, rating differences fix stimulus intensities and quantify pain discriminability with the difference in subjective pain ratings between the two stimuli. Fixing physical intensities has two practical advantages: it is easier to fix stimulus intensity than subjective perception, and the limited precision of nociceptive laser stimuli (e.g., 0.25J step size in laser stimulators) can make JND measurements less reliable. Moreover, we validated our findings with another discriminability measure, signal detection theory-derived area under the curve (AUC), which could better control nonsensory factors such as internal rating scales.

In Dataset 1, pain stimuli with higher intensity evoked greater variability in EEG responses, particularly over the central region of the brain (Fig 2A). Additionally, pain ratings were higher in the high-intensity condition than the low-intensity condition ($t(140) = 17.08$, $P < 0.0001$, Cohen's $d = 1.44$). To determine the relationship between neural variability and pain intensity discriminability (quantified as the difference of pain ratings), we then correlated them at every time point at Cz and adjusted $P$ values with false discovery rate (FDR) correction. Significant correlations were identified (i.e., time points with gray-shaded bars) across participants between neural variability difference (i.e., temporal SD difference, ΔSD) in the high- and low-intensity conditions and pain intensity discriminability in Datasets 1 (Fig 2B). This observation also received strong support from Bayesian statistics (reddish bars in Fig 2B) and region-of-interest analyses (±10 ms centered at the peak: $r = 0.43$, $P = 1.16 \times 10^{-7}$, Bayes factor [BF] = $8.05 \times 10^4$; Fig 2D).

Statistically speaking, SD is inherently correlated with mean values, which was verified in Dataset 1 (S2B Fig). A previous study has shown that mean ERP amplitude differences between high- and low-intensity conditions also correlate with pain intensity discriminability [21], which was confirmed here by applying a 100-ms sliding window analysis to the ERP waveforms (S2C and S2D Fig). ERP amplitude, thus, could potentially confound the relationship between neural variability and pain intensity discriminability described above. To rule out this possibility, we conducted control analyses. First, we performed point-by-point partial correlations between ΔSD and pain intensity discriminability controlling for the mean amplitude difference (ΔAmplitude). Significant partial correlations between ΔSD and pain intensity discriminability were observed (Fig 2C and 2D; partial $r$ around the peak = 0.45, $P = 2.23 \times 10^{-8}$, BF = $6.16 \times 10^5$). Notably, the time points showing significant partial correlations nearly perfectly overlapped with those showing simple correlations. Interestingly,

## A

### Study overview (total N = 633)

| Questions | Datasets | Results |
|---|---|---|
| ● Q1: Does neural variability replicably encode pain? | Datasets 1-5 (N = 606 HV) | Figs 2, 7 & 9 |
| ● Q2: Do neural variability and amplitude differentially encode pain? | Datasets 1-3 (N = 366 HV) | Figs 3 & 4 |
| ● Q3: Does neural variabilility reflect stimulus-general factors? | Datasets 1-3 (N = 366 HV) | Figs 5 & 6 |
| ● Q4: Is neural variability applicable to clinical data? | Dataset 6 (N = 27 PHN) | Fig 8 |

## B

### Experimental Procedure

**Datasets 1 - 3**

1 block = 40 trials

Block 1 ...... Block 2 ...... Block 3 ......

1 trial

| Fixation 3s | Stimulation 0.001~0.1s | Delay 3s | Rating 5s | Interval 1~3s |
|---|---|---|---|---|
| + | + | + | NRS 0~10 | + |

**Dataset 4 - 6**

40/60/80 trials (each intensity = 10/20 trials)

...... ......

1 trial

Stimulation    Delay    Rating    Interval

## C

### Neural variability calculation and correlation analysis

**Neural variability calculated on EEG response with sliding windows**

W1, W2, ......    Wi,    ......, WL

SD1, SD2, ......    SDi,    ......, SDL

**Neural variability averaged across trials**

**Neural variability difference between high and low intensity**

**Correlations with sensory discriminabiilty**

± 10 ms centered at the peak

Sensory discriminability

Neural variability difference    Brain topography of correlation coefficients

**Fig 1. Study overview, experimental procedures, and neural variability.** **(A)** Overview of the study. Participants in Datasets 1–5 were healthy volunteers (HV), whereas participants in Dataset 6 were chronic pain patients with postherpetic neuralgia (PHN). **(B)** Experimental procedures. All participants

in Datasets 1–3 received a total of 120 stimuli across four sensory modalities (pain, touch, audition, and vision) presented in a pseudo-random order in three blocks. After each stimulus, participants rated the perceived intensity using a numerical rating scale (NRS), which ranged from 0 (no sensation) to 10 (the strongest imaginable sensation). Participants in Datasets 4–6 received painful laser stimuli at multiple levels of intensity in a pseudo-random order. For each intensity, 10 pulses were delivered. After each stimulus, participants rated the perceived intensity using the NRS. **(C)** Calculation of temporal SD as a measure of neural variability. Temporal SD was derived from windowed EEG responses evoked by each stimulus. A series of 100 ms sliding windows with a 1 ms step was employed. SD difference (ΔSD) was then calculated by subtracting the trial-averaged SD in the low-intensity condition from that in the high-intensity condition. Correlation analyses were conducted to evaluate the relationship between neural variability and sensory intensity discriminability.

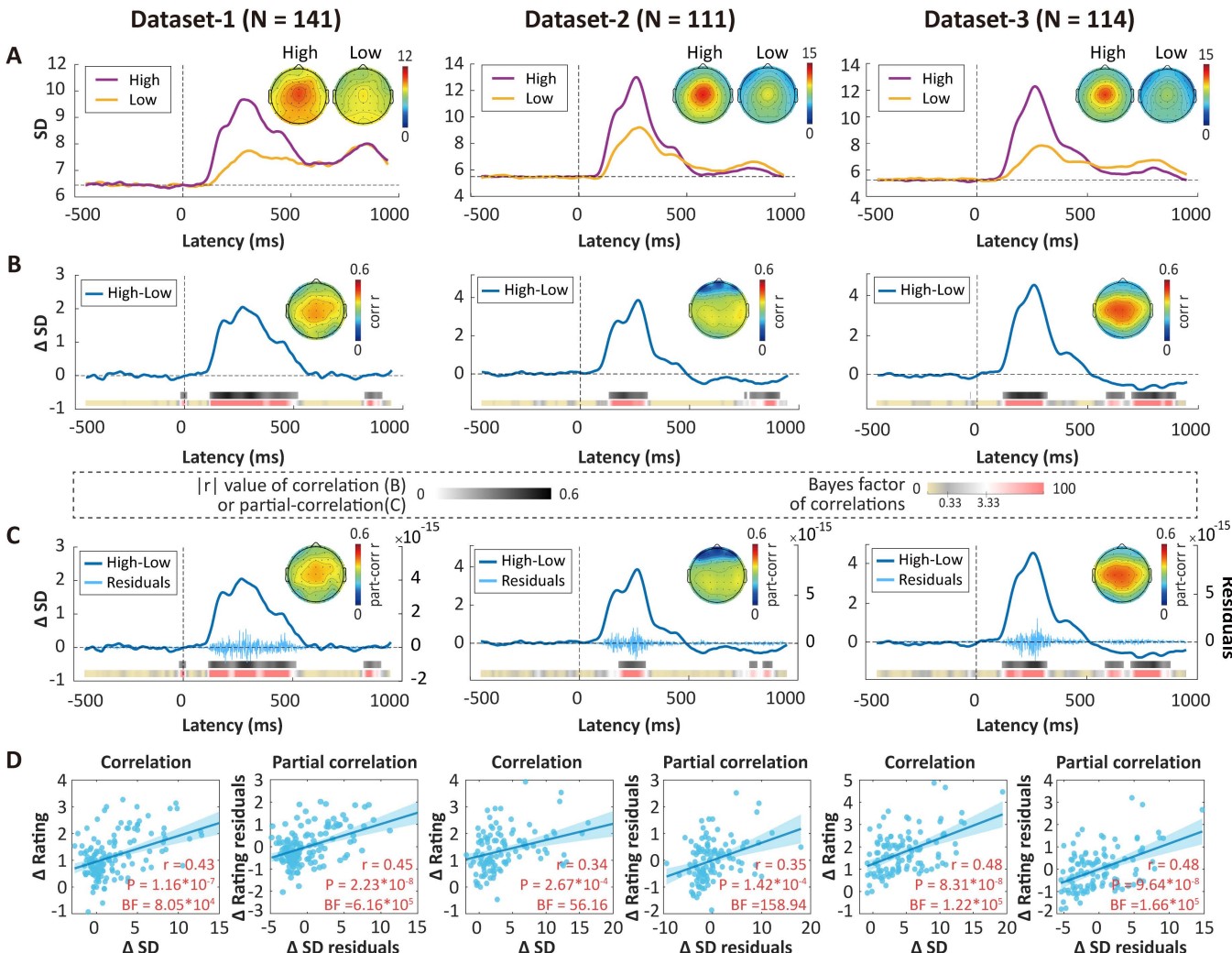

**Fig 2. Neural variability reliably correlated with pain intensity discriminability. (A)** Neural variability assessed by a variance-based measure, temporal SD of EEG responses evoked by high-intensity (violet) and low-intensity (gold) stimuli in Dataset 1 ($N = 141$), Dataset 2 ($N = 111$), and Dataset 3 ($N = 114$), respectively. **(B)** Point-by-point correlations between neural variability differences (high − low, ΔSD) and rating differences (high − low, ΔRating) in the three datasets. **(C)** Point-by-point partial correlations between ΔSD and ΔRating while controlling for amplitude differences (high − low, Δamplitude). The light-colored "bursty" curve represents the subject-averaged residuals of ΔSD after regressing out Δamplitude at each time point. **(D)** Correlations and partial correlations between values around the peak of neural variability (±10 ms) and ΔRating. Partial correlations were calculated while controlling for Δamplitude. Note that the gray bars represent Pearson's $r$ values at time points where significant correlations were observed after FDR correction. The color bars underneath display the corresponding Bayes factor values for the correlations. Error bars in D are 95% confidence intervals. The data underlying this Figure can be found in https://doi.org/10.17605/OSF.IO/QTV8A.

amplitude correlated with pain intensity discriminability at time points near N2 and P2 peaks after accounting for neural variability as well (S2E and S2F Fig), suggesting that neural variability and amplitude were mutually independent in their role in representing pain intensity discriminability. Second, we repeated the foregoing correlation analysis with an information theory-based measure of neural variability: permutation entropy (PE). PE quantifies the complexity of a time series by evaluating the ordinal relations between values in the time series rather than their absolute numerical values [42], and is thus less affected by amplitude and noise such as transient high-amplitude bursts. Significant correlations were again observed between PE difference (ΔPE) and pain intensity discriminability (S3 Fig). As a third control analysis, we subtracted trial-averaged ERPs from each trial before calculating temporal SD, that is, analyzing induced responses [43]. Still, significant correlations were observed between the neural variability difference (assessed by temporal SD and PE) of the induced responses and pain intensity discriminability in Dataset 1 (S4 Fig). Altogether, these findings lend strong support to the statement that neural variability encodes pain intensity discriminability independent of amplitude.

To test the robustness of our findings, we conducted further sensitivity analyses examining the influence of metrics for pain intensity discriminability and varying window sizes for computing neural variability. Instead of simple differences in pain ratings, we quantified pain intensity discriminability with the AUC derived from signal detection theory (SDT; for an elaborate explanation of AUC, please refer to [21]). This metric is less affected by internal rating bias and can help better address the possible influence of internal scaling of pain intensity on our findings. AUC-based pain intensity discriminability yielded highly similar results as the difference-based discriminability metric (S5 Fig), suggesting that rating bias has no substantial effect on our findings. We then calculated neural variability using alternative time window sizes of 50 and 200 ms. Our results remained consistent: pain intensity discriminability correlated with neural variability regardless of the window sizes (see S6 Fig). These findings provide evidence for the robustness of neural variability as an indicator of pain intensity discriminability.

Single-trial SD is especially susceptible to noise such as transient high-amplitude bursts and the signal-to-noise ratio could differ in high- and low-intensity conditions. To examine the effect of noise, we conducted two additional analyses. First, we computed SD using the trial-averaged EEG data, which are inherently less noisy than single-trial data and residual noise after preprocessing would be largely averaged out. Similar results to those from single-trial-based SD were obtained (S7 Fig). Furthermore, we manually added various levels of noise to single-trial EEG data during the post-stimulus period. Briefly, we sampled noise data from single-trial EEG data during the baseline period, multiplied the noise by a factor (referred to as "noise scale"), and added the resulting noise into post-stimulus EEG traces (see Noise simulation analysis in Methods for more details). Introducing additional noise to the low- or high-intensity conditions dramatically altered neural variability as measured by SD (S8A and S8C Fig). Nonetheless, noise seemed to have no substantial effects on partial correlations between pain intensity discriminability and neural variability while controlling for EEG amplitude differences (S8B and S8D Fig). Only when an excessive amount of noise (noise scale = 2) was added to the high-intensity condition did we receive no support for the partial correlation. It is noteworthy that this level of noise is implausible in real EEG signals after preprocessing. Furthermore, adding noise to the high- and low-intensity conditions simultaneously also yielded similar results (S7E and S7F Fig). Altogether, these simulation analyses suggest our findings are not artifacts caused by noise differences between intensity conditions.

To assess the replicability of our findings, we then examined the relationship between neural variability and pain intensity discriminability in two independent datasets. Positive correlations at the vertex of the scalp were also observed in both Datasets 2 and 3 (Fig 2B). Additionally, ΔSD around the peak exhibited significant correlations with pain intensity discriminability in both datasets (Dataset 2: $r = 0.34$, $P = 2.67 \times 10^{-4}$, BF = 56.16; Dataset 3: $r = 0.41$, $P = 6.88 \times 10^{-6}$, BF = 158.94; Fig 2D). Further analyses once again confirmed that ERP amplitude did not explain the correlations between neural variability and pain intensity discriminability in Datasets 2 and 3 (partial correlation in Fig 2C and 2D). Collectively, these findings underscore the replicability and reliability of neural variability as an independent indicator of pain intensity discriminability.

## Neural variability encodes pain intensity discriminability earlier and more efficiently than amplitude with distinct oscillatory profiles

Given that both neural variability and ERP amplitude have the capability to encode pain intensity discriminability, a key question emerges: how do they differ? To address this question, we first conducted dominance analysis to separate the relative contribution of neural variability and ERP amplitude to pain intensity discriminability. Dominance analysis is a method that decomposes the total determination coefficient ($R^2$) of a multiple regression model into unique $R^2$ (or "total dominance" in dominance analysis's terminology) contributed by each predictor [44,45]. We performed dominance analysis at each time point to evaluate the total dominance of neural variability and ERP amplitude. In Dataset 1, the full model with both SD and mean amplitude as predictors exhibited two peaks at around 200 and 400 ms, roughly corresponding to the N2 and P2 components of ERPs (Fig 3A). Neural variability played a greater role in the rising phase of the first peak, while ERP amplitude predominantly contributed to the second peak. Similar patterns could also be observed in Datasets 2 and 3. These findings suggest that neural variability and ERP amplitude differ in the time course of their encoding pain intensity discriminability. Specifically, neural variability plays a role equal to or greater than amplitude in encoding pain intensity discriminability around 100–300 ms after stimulus onset, while ERP amplitude plays more important roles about 300–500 ms after stimulus onset.

To further characterize the differences between neural variability and ERP amplitude, we applied a resampling approach to explore whether subject numbers influence the probability of detecting significant correlations between SD or mean amplitude and pain intensity discriminability in Datasets 1–3 (see Methods for resampling details). We found no clear evidence that either neural variability or ERP amplitude requires a smaller number of subjects to exhibit their correlations with pain intensity discriminability with 80% power (Fig 3B). Interestingly, we observed a difference in the time windows when comparing required subject numbers using SD and mean amplitude. Specifically, with the same subject number, the correlations between SD and pain intensity discriminability could be detected earlier than those between mean amplitude and pain intensity discriminability. These results further confirmed that neural variability encodes pain intensity discriminability earlier than mean amplitude.

Neural variability can be computed at the single-trial level, but ERP amplitude is generally calculated by averaging multiple trials. We thus hypothesized that another difference between neural variability and ERP amplitude is that neural variability may require a smaller number of trials to exhibit correlations with pain intensity discriminability. To test this hypothesis, we applied the resampling approach to evaluate the influence of trial numbers on the probability of detecting significant correlations between neural variability or ERP amplitude and pain intensity discriminability. The trial number that could yield a significant correlation between SD and pain intensity discriminability with 80% power was ~4 in the three datasets (Fig 3C, upper panels). In contrast, using ERP amplitude as an indicator of pain intensity discriminability required more trials (~7) to meet the same criteria in the three datasets (Fig 3C, lower panels). Therefore, neural variability encodes pain intensity discriminability more efficiently than amplitude.

To reveal the oscillatory profiles of neural variability and ERP amplitude, we band-pass filtered single-trial EEG signals according to the classical frequency bands (i.e., δ, θ, α, and β) and conducted partial correlation analysis. ERP-like deflections could be found only in the delta and theta frequency bands (Fig 4). Consistent with this observation, after controlling for neural variability, we only observed significant correlations between amplitude and pain intensity discriminability in the delta and theta frequency bands, but not in the alpha and beta frequency bands. In contrast, the partial correlations between neural variability and pain intensity discriminability after regressing out amplitude were present in all four frequency bands, even in frequencies without ERP responses (Fig 4). These findings were replicated in all three datasets (S9 Fig), suggesting that neural variability and ERP amplitude had different oscillatory profiles in encoding pain intensity discriminability.

## Neural variability potentially preferentially encodes pain intensity discriminability

After establishing the independence and replicability of the association between neural variability and pain intensity discriminability, we proceeded to investigate whether this reflects stimulus-general factors or could be potentially

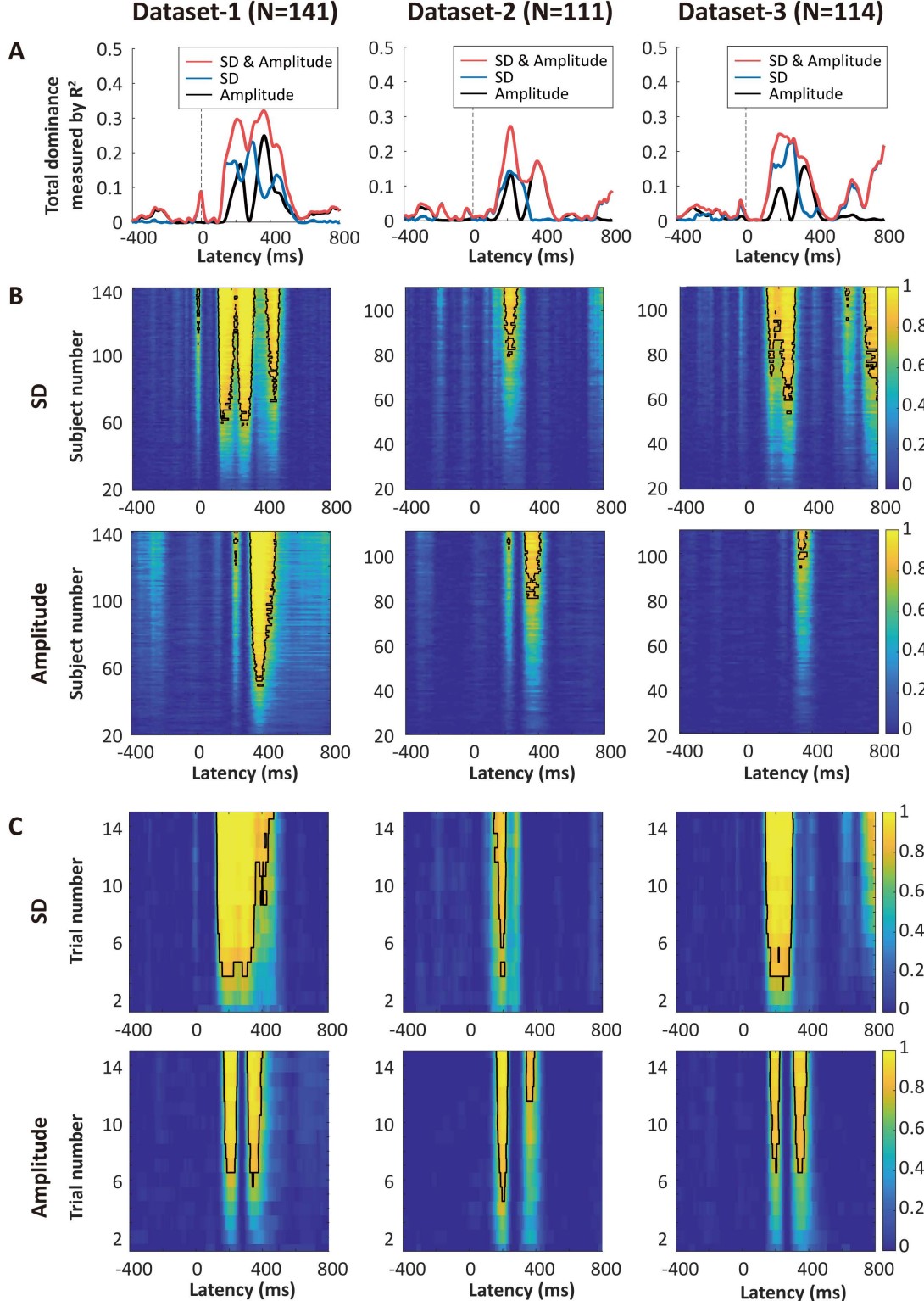

**Fig 3. Comparisons of neural variability and ERP amplitude for encoding pain intensity discriminability. (A)** Relative contributions of neural variability and ERP amplitude in Datasets 1–3. SD and ERP amplitude exhibited different contributions to pain intensity discriminability over time,

with the former having greater contributions around 100–300 ms after stimulus onset and the latter having greater contributions around 300–500 ms. Curves represent total dominance measured by determination coefficient $R^2$. **(B)** The influence of the number of subjects on the probability of detecting significant correlations between SD or ERP amplitude and pain intensity discriminability after FDR correction in the three datasets. Color bars indicate the probability of detecting the correlations. The black contour represents the probability of significance that exceeds 80%. Using neural variability or ERP amplitude as indicators of pain intensity discriminability may require a comparable number of subjects. These results further confirmed that SD and amplitude exhibited different contributions to pain intensity discriminability over time. **(C)** The influence of the number of trials on the probability of detecting significant correlations between SD or amplitude and pain intensity discriminability after FDR correction in the three datasets. Using neural variability as an indicator of pain intensity discriminability requires fewer trials than using ERP amplitude. The data underlying this Figure can be found in https://doi.org/10.17605/OSF.IO/QTV8A.

pain-preferential. Note that participants in Datasets 2 and 3 were recruited on a rolling basis and assigned to either Dataset 2 or Dataset 3 based on their pain sensitivity, which was assessed during the calibration phase. We thus combined these two datasets (denoted as Datasets 2&3) in the analyses of nonpain modality data. We found no reliable evidence for correlations between neural variability and discriminability in the tactile, auditory, or visual modalities. In Dataset 1, significant but weak correlations were observed only in the condition involving tactile stimuli within 35–180 ms, which were before the peak of ΔSD wave (see S10 Fig). However, no significant correlations were observed between the peak values of ΔSD and sensory discriminability after controlling for the mean amplitude of sensory ERPs: (1) touch: partial $r=0.13$, $P=0.14$, BF = 0.31; (2) audition: partial $r=0.14$, $P=0.09$, BF = 0.42; (3) vision: partial $r=0.05$, $P=0.59$, BF = 0.12 (Fig 5A–5C). Direct comparisons between pain and nonpain modalities confirmed that neural variability correlated more with pain intensity discriminability than nonpain intensity discriminability: (1) pain versus touch: $z=2.95$, $P=0.003$; (2) pain versus audition: $z=2.91$, $P=0.004$; (3) pain versus vision: $z=3.68$, $P=0.0002$. These results were replicated in Datasets 2&3. There was evidence against correlations between neural variability and sensory discriminability in nonpain modalities: (1) touch: partial $r=0.06$, $P=0.35$, BF = 0.13; (2) audition: partial $r=0.09$, $P=0.20$, BF = 0.19; (3) vision: partial $r=0.09$, $P=0.18$, BF = 0.21 (Fig 5D–5F). Direct comparisons between pain and nonpain modalities showed stronger correlations between neural variability and pain intensity discriminability: (1) pain versus touch: $z=4.12$, $P=3.7 \times 10^{-5}$; (2) pain versus audition: $z=3.96$, $P=7.4 \times 10^{-5}$; (3) pain versus vision: $z=3.86$, $P=0.0001$.

The results above offer preliminary evidence that neural variability might preferentially encode pain intensity discriminability. However, a potential confounding factor arises from the possibility that sensory discriminability, quantified as rating differences, could vary across modalities. Indeed, the rating disparities for painful stimuli were notably smaller compared to those observed for auditory and visual stimuli (e.g., paired-sample $t$ tests in Dataset 1: pain versus audition: $t(140) = -11.45$, $P=7.97 \times 10^{-22}$; pain versus vision: $t(140) = -10.31$, $P=6.75 \times 10^{-19}$; Datasets 2&3: pain versus audition: $t(224) = -27.90$, $P=7.98 \times 10^{-75}$; pain versus vision: $t(224) = -14.47$, $P=6.12 \times 10^{-34}$). A significant difference was also observed between painful and tactile stimuli in Datasets 2&3 ($t(224) = 6.08$, $P=5.09 \times 10^{-9}$). To ensure rigorous and fair comparisons, we implemented a rating matching procedure (see Methods for details). This approach enabled us to identify subgroups of participants who exhibited comparable intensity ratings across pairs of sensory modalities, specifically pain versus touch, pain versus audition, and pain versus vision. Through this method, Dataset 1 yielded 70 participants for pain-touch matching, 33 for pain-audition matching, and 58 for pain-vision matching. Within these matched subgroups, significant correlations between neural variability and pain intensity discriminability were consistently observed even after controlling for the mean amplitude of respective ERPs, while no significant correlations were found in nonpain modalities: (1) pain versus touch: pain, partial $r=0.49$, $P=2.27 \times 10^{-5}$, BF = $1.10 \times 10^3$; touch, partial $r=0.17$, $P=0.17$, BF = 0.36; (2) pain versus audition: pain, partial $r=0.40$, $P=0.02$, BF = 2.21; audition, partial $r=0.17$, $P=0.35$, BF = 0.32; (3) pain versus vision: pain, partial $r=0.49$, $P=9.10 \times 10^{-5}$, BF = 303.34; vision, partial $r=-0.02$, $P=0.87$, BF = 0.16 (Fig 6A–6C). Direct comparisons showed similar findings: (1) pain versus touch: $z=2.08$, $P=0.037$; (2) pain versus audition: $z=0.96$, $P=0.34$; (3) pain versus vision: $z=2.93$, $P=0.003$. Consistent correlational findings were obtained in Datasets 2&3: (1) pain versus touch: pain, partial $r=0.54$, $P=9.07 \times 10^{-9}$, BF = $1.71 \times 10^6$; touch, partial $r=0.14$, $P=0.16$, BF = 0.33; (2) pain versus

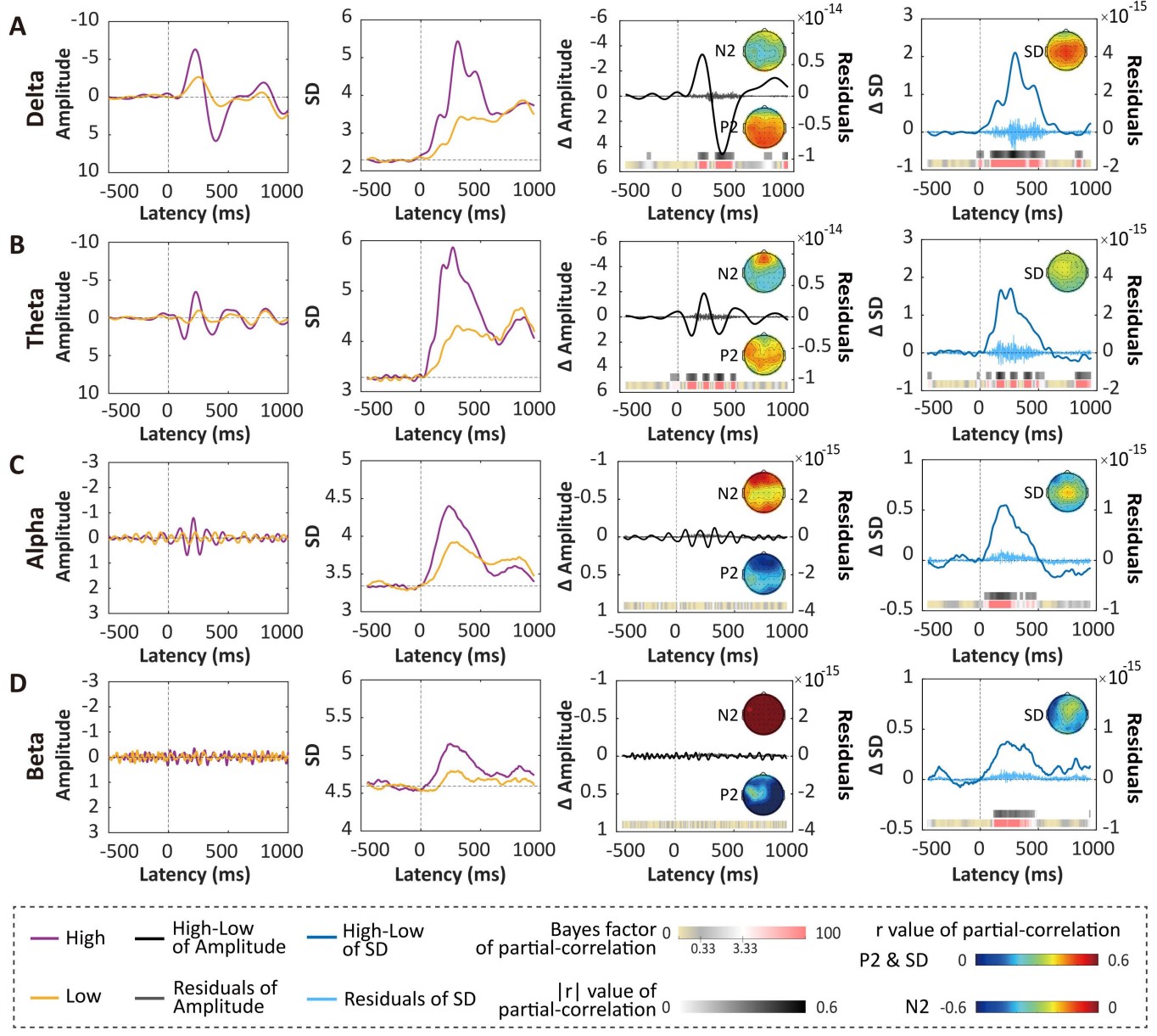

**Fig 4. Comparisons of neural variability and ERP amplitude at different frequency bands for encoding pain intensity discriminability in Dataset 1.** (A–D) ERP amplitude, neural variability, and the differential values between high-intensity and low-intensity stimuli at the delta (A, 1–4 Hz), theta (B, 4–8 Hz), alpha (C, 8–12 Hz), and beta (D, 12–30 Hz) bands, and their partial correlations while controlling for each other with pain intensity discriminability. The two left panels show ERP amplitude and neural variability of band-pass filtered time series averaged across trials evoked by high- (violet) and low-intensity (gold) stimuli. The two right panels show point-by-point partial correlations of amplitude differences (high − low, ΔAmplitude) and SD differences (high − low, ΔSD) with pain intensity discriminability, respectively. Note that the gray bars represent Pearson's *r* values at time points where significant correlations were observed after FDR correction. The color bars underneath display the corresponding Bayes factor values for the correlations. The topographies represent partial *r* values within a ±10 ms window around the peak. For ΔAmplitude calculations in A and B, the latencies of N2 and P2 peaks were utilized, while for C and D, where clear ERPs were not observed, latencies corresponding to the minimal and maximal values within the 100–500 ms time window were used. The data underlying this Figure can be found in https://doi.org/10.17605/OSF.IO/QTV8A.

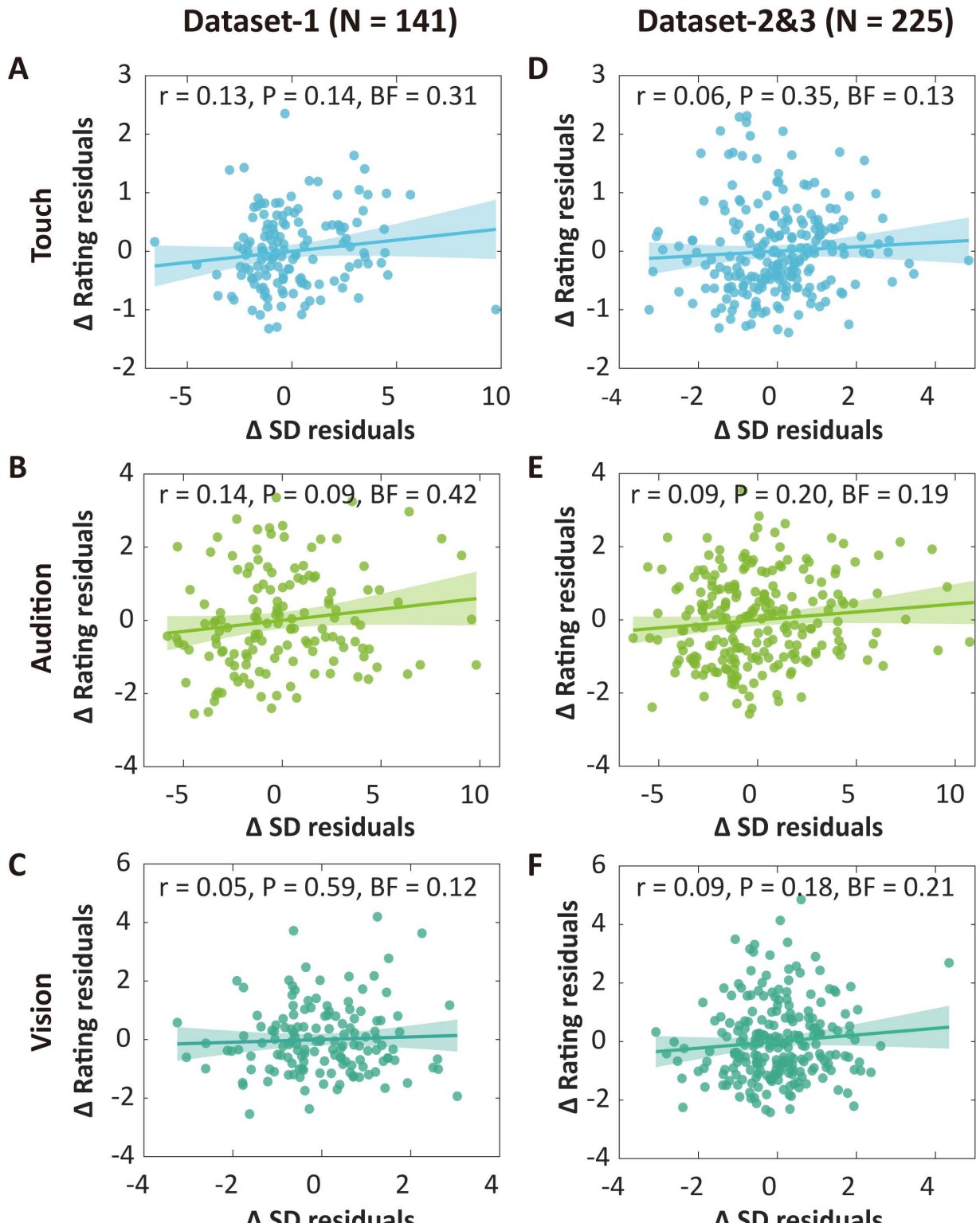

**Fig 5. Neural variability of EEG responses evoked by tactile, auditory, and visual stimuli did not reflect their respective sensory intensity discriminability.** (A–C) Partial correlations between neural variability and sensory intensity discriminability of tactile, auditory, and visual stimuli in Dataset 1. See point-by-point partial correlations in S10 Fig. No significance was observed for all nonpain modalities in partial correlations while controlling for

mean amplitude differences of ERPs. (D–F) Partial correlations between neural variability and sensory intensity discriminability in Datasets 2&3. No significant partial correlations while controlling for mean amplitude differences were observed for any of these sensory modalities. The $r$ shown in panels represents $r$ values of partial correlation. BF is short for Bayes factor. The data underlying this Figure can be found in https://doi.org/10.17605/OSF.IO/QTV8A.

audition: pain, partial $r=0.55$, $P=3.85\times10^{-4}$, BF = 103.77; audition, partial $r=0.10$, $P=0.56$, BF = 0.22; (3) pain versus vision: pain, partial $r=0.54$, $P=1.51\times10^{-7}$, BF = $1.23\times10^5$; vision, partial $r= -0.03$, $P=0.77$, BF = 0.14 (Fig 6D–6F). Direct between-modality comparisons led to confirmatory results as well: (1) pain versus touch: $z=3.18$, $P=0.001$; (2) pain versus audition: $z=2.16$, $P=0.031$; (3) pain versus vision: $z=4.01$, $P=6.1\times10^{-5}$. These consistent results demonstrate that neural variability may not reflect stimulus-general factors, but possibly preferentially encodes pain intensity discriminability.

**Generalizability and preliminary clinical relevance of neural variability as an indicator of pain intensity discriminability**

In Datasets 1–3, the difference of laser stimulus energy between high- and low-intensity conditions remained fixed at 0.5 J. To examine the generalizability and potential clinical relevance of the correlations between neural variability and pain intensity discriminability, we conducted additional analyses on Datasets 4–6, in which participants received laser stimuli of multiple intensities, specifically, 2.5, 3.0, 3.5, and 4.0 J in Dataset 4 ($N=96$, healthy volunteers) and Dataset 5 ($N=144$, healthy volunteers), and 3, 3.25, 3.5, 3.75, 4, and 4.25 J in Dataset 6 ($N=27$, patients with postherpetic neuralgia [PHN]). Note that PHN patients received stimulation both on the shingles-affected skin area (pain-affected) and mirrored skin area on the contralateral side (pain-unaffected). We did not directly compare patients and healthy volunteers due to differences in datasets and demographics between these groups. Instead, we examined within-individual differences between affected and unaffected sides in patients.

In Dataset 4, pain ratings were significantly different across stimulus intensities (one-way repeated measures ANOVA, $F(3, 380) = 182.47$, $P=2.86\times10^{-73}$; Fig 7A). Temporal SDs were significantly different within $100\sim500$ ms in all pairs of high- and low-intensity stimuli (Fig 7A). Correlations between ΔSD and pain intensity discriminability showed significance in four pairs of high- and low-stimulus intensities as illustrated in Fig 7B. To assess the reliability of the above results, we replicated the analysis using a published dataset (Dataset 5). In this dataset, pain ratings were also significantly different across stimulus intensities (one-way repeated measures ANOVA, $F(3, 572) = 100.95$, $P=1.88\times10^{-52}$; Fig 7C). Temporal SDs were significantly different within $100\sim500$ ms in all pairs of high- and low-stimulus intensities except the pair of 3.0 J versus 2.5 J (Fig 7C). Correlations between ΔSD and pain intensity discriminability were significant in all pairs of high- and low-intensity stimuli (Fig 7D).

In Dataset 6, stimulus intensity had significant influence and stimulation side had no significant influence on subjective pain ratings, and their interaction was not significant (two-way repeated measures ANOVA, stimulus intensity: $F(5, 312) = 5.46$, $P=7.85\times10^{-5}$; side: $F(1, 312) = 0.28$, $P=0.60$; interaction: $F(5, 312) = 0.30$, $P=0.92$). Significant differences were found between most pairs of stimulus intensities with a difference exceeding 0.5 J on both sides (Fig 8A), which also showed significant temporal SD differences (Fig 8A). Due to limited number of subjects, we performed Spearman's correlation analyses between ranked ΔSD and ΔRating. Significant correlations were observed in four pairs of high- and low-stimulus intensities on the unaffected side and zero pairs on the affected side (Fig 8B and 8C). The associations between neural variability and pain intensity discriminability, therefore, seem to be disrupted to some extent on the neuropathic pain-affected side.

**Neural variability reflects general intraindividual sensory perception, but not interindividual sensory sensitivity**

To investigate the relationship more comprehensively between neural variability and pain, we then examined whether neural variability reflects interindividual pain sensitivity and intraindividual pain perception. For interindividual sensitivity, temporal SD waves were averaged across all trials regardless of stimulus intensity, and then correlated with

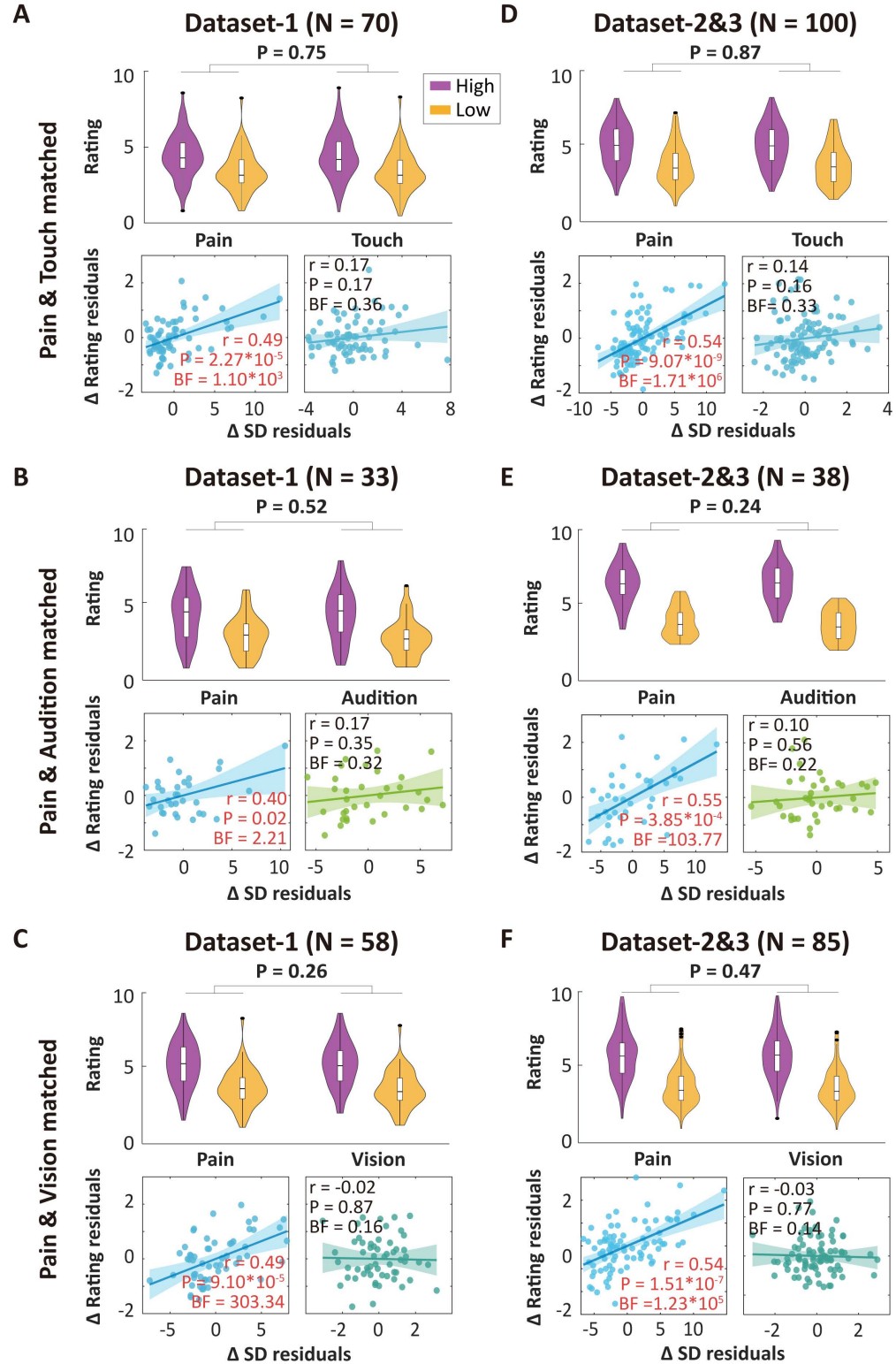

**Fig 6. Preferentiality of neural variability as an indicator of pain intensity discriminability by matching the ratings between pain and non-pain modalities. (A** and **D)** The upper panels show distributions of ratings for high-intensity and low-intensity stimuli matched between pain and tactile modalities in Datasets 1 and 2&3. The high-low rating differences between pain and touch were not significant after rating matching. The lower panels

show partial correlations between SD differences and rating differences for the matched subset. (B–C, E–F) Similar figures for subsets matched between pain and auditory modalities, and between pain and visual modalities in Datasets 1 and 2&3. Significant partial correlations were found in all matched subsets for pain intensity discriminability, but not for tactile, auditory, and visual modalities in both datasets. The r shown in scatterplots represents r values of partial correlation. BF is short for Bayes factor. Error bars are 95% confidence intervals. The data underlying this Figure can be found in https://doi.org/10.17605/OSF.IO/QTV8A.

trial-averaged ratings. After controlling for the mean ERP amplitude, no significant correlations were found between trial-averaged SD and ratings for all modalities in Dataset 1 and Datasets 2&3 (Fig 9A and 9C). For intraindividual sensory perception, correlation analyses were conducted at the single-trial level for each subject, and the Fisher $z$-transformed correlation coefficients were then compared with zero using a one-sample $t$ test. In contrast to the results for interindividual sensitivity, significant correlations were observed for all modalities (Fig 9B and 9D). These findings align with previous studies demonstrating that the amplitude of most brain responses to nociceptive stimuli, though nonselective, reflects pain perception at the intraindividual level, while ERPs per se do not encode pain sensitivity at the interindividual level [15].

## Discussion

Moving beyond the traditional amplitude-based approach, we systematically investigated the relationship between neural variability and pain, particularly individual differences in pain intensity discriminability, in six large EEG datasets (total $N = 633$). Five major findings were obtained. First, there were robust and replicable correlations between neural variability and pain intensity discriminability across individuals in multiple datasets. Second, almost no significant correlations were observed between neural variability and sensory discriminability in tactile, auditory, and visual modalities, even when the perceptual ratings of nonpain modalities were matched with those of pain. Third, neural variability and ERP amplitude were mutually independent in encoding pain intensity discriminability, and they had distinct temporal dynamics, efficiency, and underlying oscillatory profiles. Fourth, the association between neural variability and pain intensity discriminability appeared to be partly disrupted on the affected side of PHN patients. Fifth, neural variability correlated with perceptual ratings in pain and nonpain modalities at the intraindividual level but did not significantly correlate with individual differences in sensory sensitivity. Taken together, we demonstrate that neural variability is a replicable and potentially preferential indicator of pain intensity discriminability, thereby enhancing the understanding of neural encoding of pain intensity discriminability and facilitating a more comprehensive analysis of neural activity incorporating both neural variability and ERP amplitude.

ERP amplitude is a central metric in many EEG studies [5,15,21,46,47]. However, neural activity is highly variable across time and such variability cannot be captured by the averaged amplitude itself [6,8,48]. Importantly, recent studies have linked neural variability to multiple cognitive processes including perception and sensory discrimination [9,49,50]. Inspired by these discoveries, we moved beyond the traditional focus on amplitude of neural activity and investigated how temporal neural variability of EEG responses relates to pain, in particular to interindividual pain intensity discriminability. Consistent with emerging studies showing a correlation between neural variability and pain [32,33], we found that neural variability reliably correlated with pain intensity discriminability in multiple large datasets. This association remained robust across different measures of pain intensity discriminability (i.e., rating differences and AUC) and different window sizes (i.e., 50 ms, 100 ms, and 200 ms). This replicable association between neural variability and pain intensity discriminability parallels the relationship observed between ERP amplitude and pain intensity discriminability [20,21]. Moreover, similar to ERP amplitude, neural variability did not significantly correlate with pain sensitivity across individuals, but correlated with intraindividual perceptual ratings across trials [15]. These findings highlight that neural variability and ERP amplitude could have similar roles in encoding pain and suggest that neural variability and ERP amplitude should both be considered in pain studies.

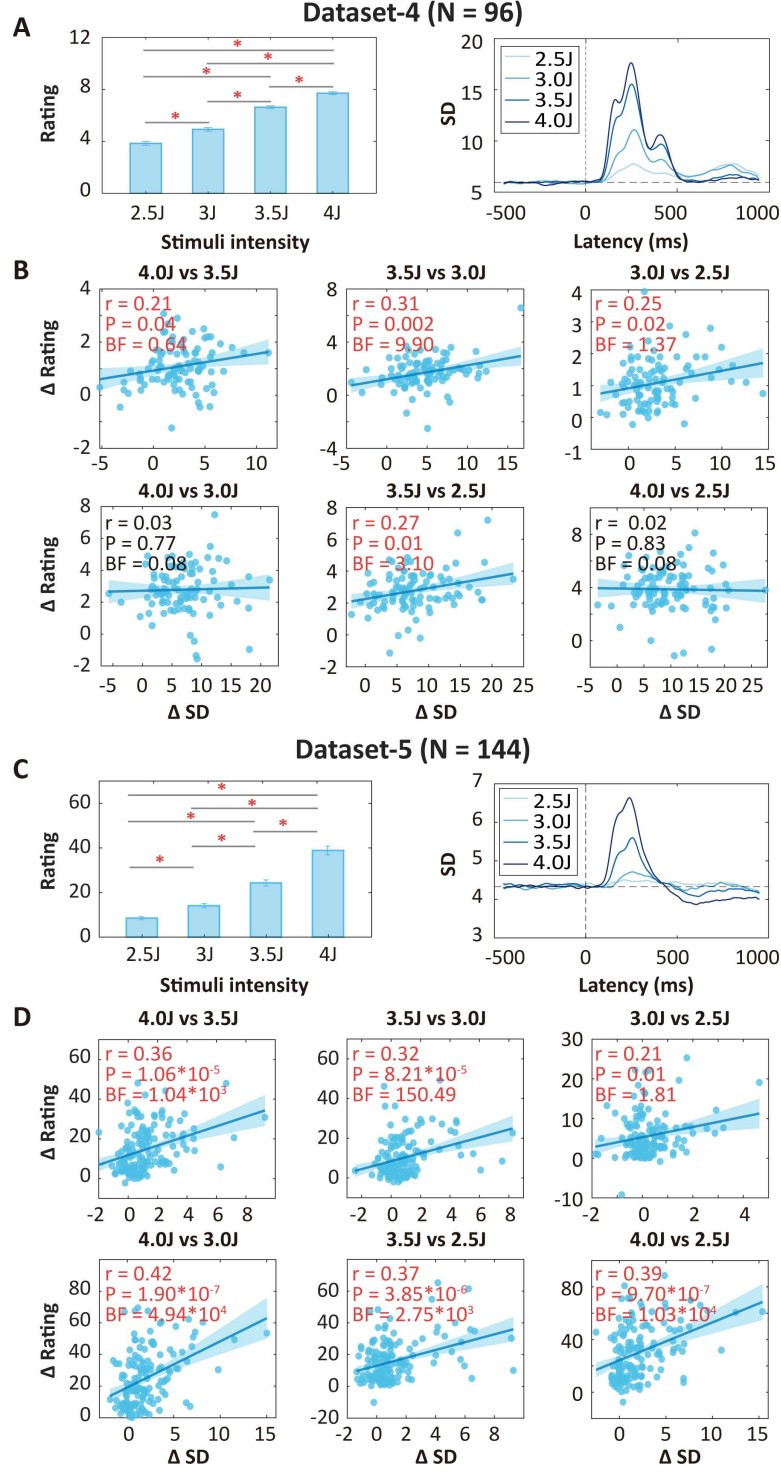

**Fig 7. Neural variability as an indicator of pain intensity discriminability in Datasets 4 and 5 with multiple levels of nociceptive laser stimuli.**
**(A** and **C)** Pain intensity ratings (left panel) and temporal SD (right panel) in Dataset 4 (A: healthy subjects, $N = 96$) and Dataset 5 (C: healthy subjects, $N = 144$). Notably, ratings were significantly different across stimulus intensities. (**B** and **D**) Correlations between ΔSD around the peak and ΔRating in all pairs of stimulus intensities. For Dataset 4 (B), significant correlations were found in discriminating between intensities with a 0.5 J difference (upper panels), as well as between 3.5 and 2.5 J (lower middle panel). For Dataset 5 (D), significant correlations were found in all pairs of stimulus intensities.

BF is short for Bayes factor. Error bars in the left panels of A and C are standard errors of means, and error bars in B and D represent 95% confidence intervals. The data underlying this Figure can be found in https://doi.org/10.17605/OSF.IO/QTV8A.

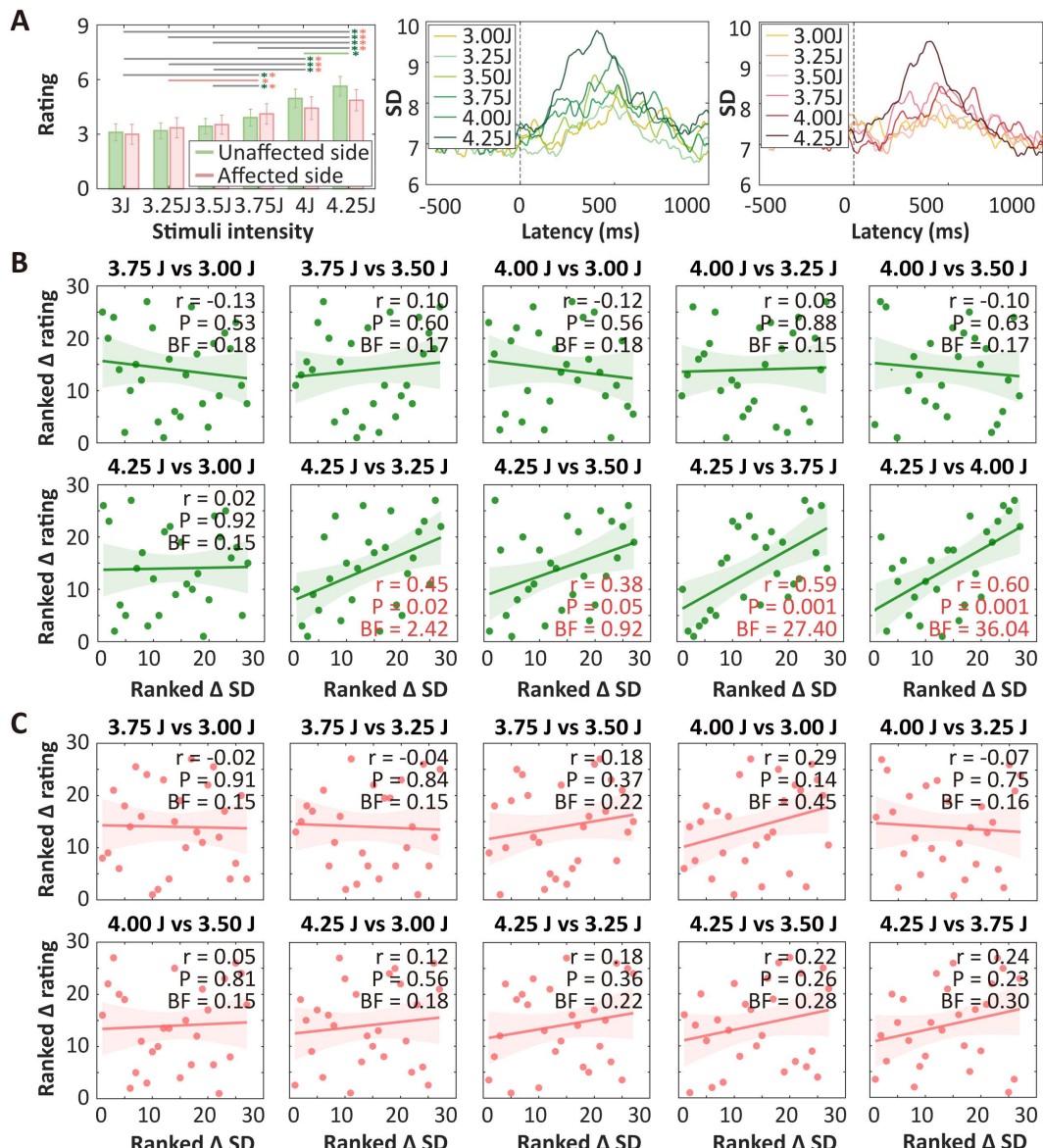

**Fig 8. Neural variability as an indicator of pain intensity discriminability in Dataset 6 (chronic pain patients) with multiple levels of laser stimuli. (A)** Pain intensity ratings (left panel), temporal SD in the unaffected and affected sides (middle and right panels) in Dataset 6 (PHN patients, $N = 27$). **(B and C)** Correlations between ΔSD and pain intensity discriminability were assessed for both the unaffected side (B) and affected side (C). Significant correlations were observed in four pairs of intensities on the unaffected side and no pairs of intensities on the affected side, indicating a disruption of the correlations in the affected side. BF is short for Bayes factor. Error bars in A are standard errors of means, and error bars in B and C represent 95% confidence intervals. The data underlying this Figure can be found in https://doi.org/10.17605/OSF.IO/QTV8A.

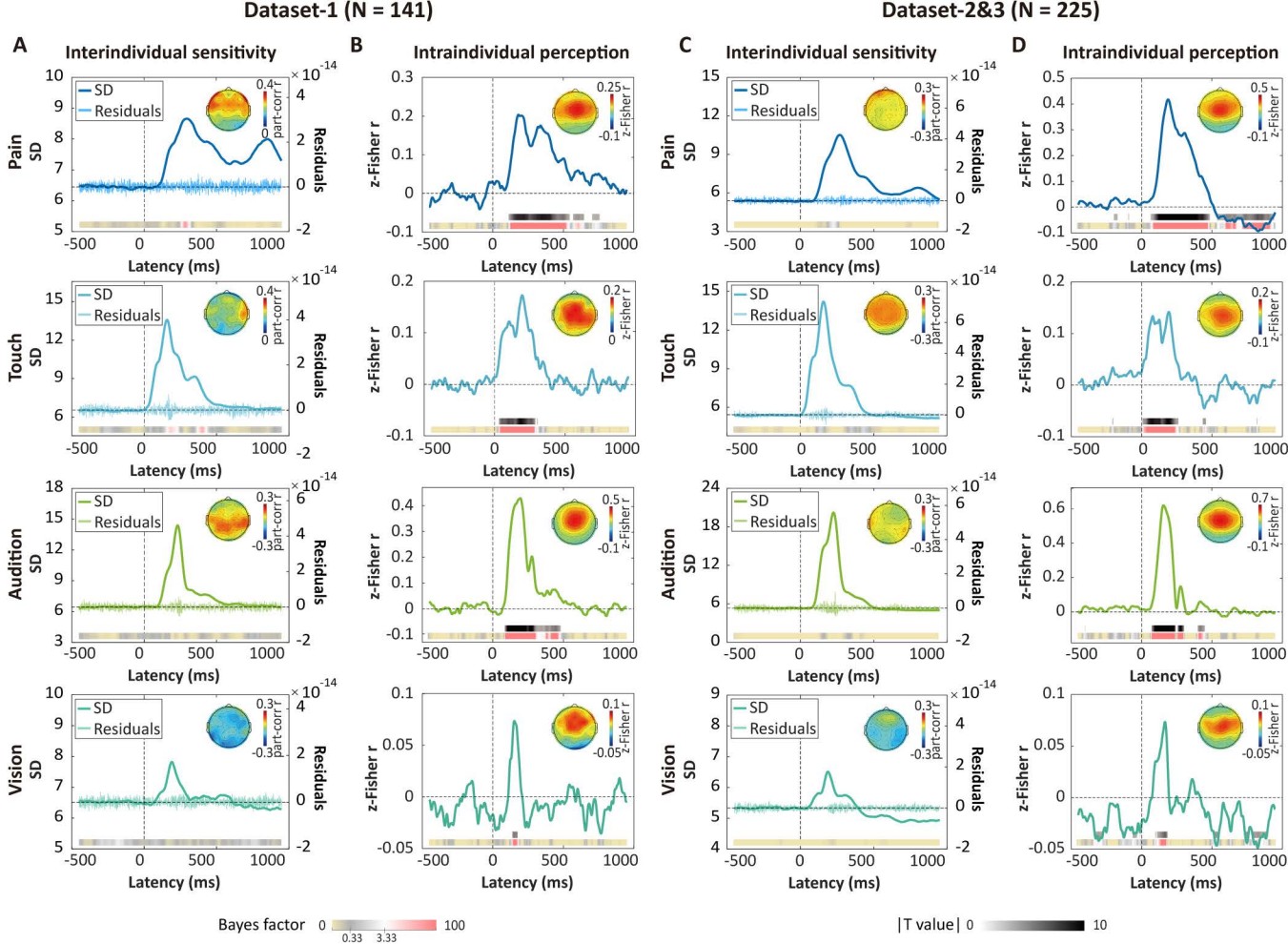

**Fig 9. Neural variability does not encode interindividual sensory sensitivity but reflects intraindividual sensory perception. (A)** Point-by-point partial correlations between trial-averaged SD ([High + Low]/2) and sensitivity measured by mean ratings ([High + Low]/2) to sensory stimuli of four modalities in Dataset 1 ($N=141$). No significant correlations were found in any modality. Note that no gray bars are shown because none of the partial correlation coefficients survived FDR correction (all $P$ values > 0.05). The color bars display the corresponding Bayes factor values for the correlations. Note that part-corr is short for partial correlation. The light-colored "bursty" curves represent the subject-averaged residuals of mean SD after regressing out mean amplitude. **(B)** Point-by-point intraindividual correlations between neural variability and ratings to sensory stimuli for each trial in Dataset 1 ($N=141$). The gray bars represent t-statistics at time points where significant differences were observed after FDR correction. **(C)** Point-by-point partial correlations between trial-averaged SD and mean ratings to sensory stimuli of four modalities in Datasets 2&3 ($N=225$). **(D)** Point-by-point intraindividual correlations between neural variability and ratings to sensory stimuli for each trial in Datasets 2&3 ($N=225$). These results illustrate that neural variability encodes sensory perception at the intraindividual level, but does not encode sensory sensitivity at the interindividual level. The data underlying this Figure can be found in https://doi.org/10.17605/OSF.IO/QTV8A.

SD and mean values are associated mathematically. One could thus argue that the observed correlations between neural variability (measured by temporal SD) and pain may be simply a manifestation of the underlying relationships between ERP amplitude (measured by mean) and pain [15,21]. However, several lines of evidence speak against this notion. First, we observed partial correlations between neural variability and pain intensity discriminability after controlling for ERP amplitude. Second, we found significant correlations between pain intensity discriminability and another information theory-based measure of neural variability, namely, PE, which is less influenced by the absolute values of ERP amplitude [42]. Third, the correlations between neural variability and pain intensity discriminability remained in induced

EEG responses. Since the mean amplitude of induced signals has no variance (i.e., having a constant value of 0), it would not confound the correlation between EEG variability and discriminability. Fourth, simulation analyses demonstrated that our findings were robust across a variety of noise levels, indicating that signal-to-noise ratio differences between intensity conditions could not explain the correlation between neural variability and pain intensity discriminability.

Importantly, we also found differential roles and oscillatory profiles of neural variability and ERP amplitude as indicators of pain intensity discriminability. Neural variability accounted for pain intensity discriminability equally well or even better than ERP amplitude in the early time window following stimulus onset, while ERP amplitude proved more crucial in the later window. This observation may be explained by SD's sensitivity to small fluctuations in neural activity compared to means. Fewer trials were also needed to establish significant correlations between neural variability and pain intensity discriminability. Neural variability thus seems more efficient in encoding pain intensity discriminability, possibly because it can be measured relatively accurately at the single-trial level. Furthermore, distinct oscillatory profiles were found for the roles of neural variability and ERP amplitude in encoding pain intensity discriminability. While ERP amplitude was mainly observed in delta and theta frequencies, neural variability involved activity in broader frequency ranges, which agrees with previous studies showing that neural variability may be a consequence of oscillations in alpha-beta bands [51]. Taken together, these findings provide further evidence that the relationship between neural variability and pain intensity discriminability cannot be attributed to ERP amplitude. Instead, neural variability constitutes an independent and critical aspect of EEG responses with distinct mechanisms. These findings are in accord with previous research demonstrating the independent contributions of variability and mean amplitude to cognitive processes [52–54].

The independent and different roles of neural variability and ERP amplitude underscore the notion that the "noise" of neural activity is not purely noise, but functionally informative [55]. Single-trial EEG typically appears very "noisy", varying dramatically across time and even suppressing ERP responses [56,57]. This temporal variability is treated as noise and averaged out in the traditional trial averaging procedure [58]. However, our findings show that this fluctuating temporal dynamics contain information about pain intensity discriminability. This point is illustrated well by the "bursty" residual time series of neural variability after regressing out amplitude: the residual neural variability differed in the high- and low-pain conditions and correlated with pain intensity discriminability. Our findings are consistent with previous studies associating task-related neural variability with sensory perception. For example, neural variability has been correlated with visual contrast discrimination threshold [49], and variability of fMRI blood-oxygenation-level–dependent (BOLD) signals has shown correlations with discriminability in touch and visual motion [9,49,50]. Notably, most previous task-based studies examine trial-by-trial neural variability, that is, how neural activity varies across trials [49,51,59]. We, on the other hand, focused on temporal neural variability—moment-to-moment fluctuations in neural activity—which has largely been examined in the context of resting-state research [6]. Our study thus extends this line of research by demonstrating that moment-to-moment neural variability responds to external stimuli and is related to pain within task-based paradigms.

Why does temporal neural variability encode pain intensity discriminability? One possibility is that neural variability reflects the precision of pain predictions. Pain perception can be theorized as a Bayesian inference process, which posits that perceived pain is the combination of bottom-up nociceptive inputs and top-down pain predictions weighted by their relative precision [60]. This framework has received increasing empirical support across pain and other sensory modalities [1,61–63]. From this perspective, neural variability may reflect perceptual uncertainty or shifts in attentional weighting in the prediction of pain. Consequently, differential neural variability in the high- and low-intensity conditions may signal varying precision of predictions in different conditions, which then results in change in pain intensity discriminability. This interpretation aligns with previous studies linking neural variability to perceptual uncertainty [64–66] and recent work demonstrating correlations between moment-to-moment variability of BOLD signals and perceptual precision in a learning task [66]. Although uncertainty in pain predictions has been associated with nociceptive-evoked ERP responses [62], there remains a lack of pain research directly investigating the relationship between pain prediction precision and neural

variability. Future studies are in need to directly test the role of pain prediction precision in the association between neural variability and pain intensity discrimination.

Interestingly, we also observed that the association between neural variability and pain intensity discriminability seemed not stimulus-general, but potentially pain-preferential. Painful and nonpainful stimuli have been shown to evoke similar ERP responses [19,39], and all sensory stimuli are salient and attention-grabbing [67], leading to the influential theory that ERP responses evoked by transient stimuli reflect salience. However, our rating-matching approach provides some indirect evidence against this salience-based interpretation. Intensity ratings have been shown to highly correlate with stimulus salience (e.g., $r = 0.84$ in [68] and $r = 0.93$ in [69]). Accordingly, stimulus salience could be more or less similar for painful and nonpainful stimuli after intensity rating matching. Given that neural variability had stronger correlations with pain intensity discriminability after intensity rating matching, this association seems not stimulus-general, and salience might not have substantially affected our findings. However, painful stimuli may command greater attention than equally intense nonpainful stimuli [70,71]. Since no salience ratings or physiological signals of arousal like skin conductance responses were collected [69], it is impossible to directly test the influence of salience on our findings and rule out this alternative interpretation. Further research needs to directly examine the role of attention in our findings. If pain preferentiality is confirmed after directly controlling for attention, this phenomenon could reflect fundamentally different mechanisms underlying intensity discriminability of pain compared with nonpain modalities. While the present study only provides preliminary evidence for preferential encoding of pain discriminability, the informational value of pain intensity may explain the potential pain preferentiality. Intensity discriminability seems to play a more functionally important role in pain than in nonpain modalities. Intensity is a dominant part of sensory information which individuals extract from painful stimuli [21]. In contrast, touch, audition, and vision encode plentiful information other than intensity, for example, texture and humidity for touch, timbre and pitch for audition, color and shape for vision. The informational value of pain intensity may thus warrant the nervous system to selectively encode this information. This interpretation is, however, speculative and needs to be tested with carefully designed experiments in the future.

In addition to the theoretical value of introducing temporal neural variability as an indicator of pain intensity discriminability, the generalizability and preliminary clinical implications of our study are noteworthy. In Datasets 4 and 5, which included stimuli with multiple intensity levels, we also observed correlations between neural variability and pain intensity discriminability. In contrast, Dataset 6—comprising PHN patients with a smaller sample size—showed fewer pairs of stimulus intensities with significant correlations. While this discrepancy may partly stem from the limited sample size, it could also reflect an aging effect, as patients in Dataset 6 were older than the healthy volunteers in Datasets 4 and 5, and older adults generally demonstrate poor pain intensity discriminability [72,73]. Interestingly, the pain-affected side had a worse performance in discriminating the same pair of stimuli. The associations between neural variability and pain intensity discriminability, therefore, seem to be disrupted on the neuropathic pain-affected side. Previous studies have also revealed that patients with chronic pain exhibit abnormal pain intensity discriminability, which correlates with the effectiveness of their treatment [22,23,25,26,74,75]. While consistent with these previous studies in principle, our findings should be interpreted with caution due to the limited sample size. They cannot provide immediate insights into chronic pain mechanisms or clinical utility. Rather, they only lend preliminary support to the potential clinical relevance of neural variability as an indicator of pain intensity discriminability. Future studies need to validate our findings in different types of chronic pain and in studies with larger clinical sample sizes.

Our study has some limitations. First, we have only focused on pain intensity discriminability. Painful stimuli can differ in various ways, including their spatial locations or temporal sequences. While intensity has long been the center of pain research, some studies have found distinct mechanisms of pain intensity and other aspects of pain such as location [76]. It is of interest to explore whether neural variability also encodes spatial discrimination of pain. Second, although replicable and potentially preferential correlations were found between neural variability and pain intensity discriminability, the causality of this relationship remains unclear. It is likely that neural variability merely reflects, rather than determines pain

intensity discriminability. Further investigations employing broader techniques, such as neurophysiology and neuromodulation, are needed to examine the causal direction [77,78]. Third, the current study used only one type of pain, namely laser heat pain. Previous studies have highlighted the potential disparities in neural mechanisms across different types of pain [79]. It is thus essential to explore whether neural variability encodes the discriminability of alternative forms of pain, such as mechanical pain or chemical pain. Fourth, the clinical implications of our findings need more testing in patient groups. While we did test the ability of neural variability to serve as an indicator of pain intensity discriminability in Dataset 6, it is only a small clinical dataset with only one type of chronic pain. Without broader testing in patients, the clinical relevance of neural variability is still limited. Addressing these limitations, future studies will deepen the understanding of how neural variability relates to pain and facilitate valuable clinical applications of neural variability.

## Methods

### Datasets and participants

Six EEG datasets were utilized in this study [15,21,80,81]. Datasets 1–5 comprised signals recorded from 606 healthy volunteers who were pain-free and had no history of chronic pain, neurological disorders or psychiatric disorders. Dataset 6 was collected from 29 patients with PHN (2 were excluded in this study due to insufficient trials for calculating the neural variability difference). Dataset 1 was recorded using the BioSemi EEG system, while Datasets 2–6 were recorded using the Brain Products EEG system. The detailed demographics are as follows: (1) Dataset 1 has 141 participants (54 males) aged $21.8 \pm 4.7$ years (mean $\pm$ SD). (2) Dataset 2 has 111 participants (54 males) aged $20.9 \pm 2.3$ years. (3) Dataset 3 has 114 participants (40 males) aged $20.7 \pm 2.3$ years. (4) Dataset 4 has 96 participants (51 males) aged $21.6 \pm 1.7$ years. (5) Dataset 5 has 159 participants in total. These participants underwent two sessions of experiments. Here we report results from the first session, but similar findings could be obtained in the second session as well. Fifteen participants were removed during preprocessing (see EEG recording and preprocessing below), leaving 144 participants (70 males) aged $39.4 \pm 17.9$ years. (6) Dataset 6 has 27 participants (12 males) aged $61.7 \pm 11.1$ years.

Note that we did not pool all datasets, but instead analyzed different datasets separately. In other words, each analysis was conducted on a homogeneous dataset. Dataset heterogeneity thus would not only have no effect on our analyses, but also ensure our findings' generalizability across various conditions such as EEG recording systems, sample sizes, and age groups.

### Ethics statement

All participants gave written informed consent and were paid for their participation. The experimental procedures were approved by the local ethics committee at the Chinese Academy of Sciences (Datasets 1–3; reference no. H17025) and Southwest University (Datasets 4 and 6; reference no. H15019). Dataset 5 was approved by the Ethics Committee of the Medical Faculty of the Technical University of Munich (reference no. 512/19 S-SR). These experiments were conducted in accordance with the Declaration of Helsinki.

### Experiments

Experimental procedures were identical in Datasets 1–3 (shown in Fig 1B). Participants were seated in a comfortable position within a dimly lit, quiet, and temperature-regulated environment. The entire experiment had three blocks, with each block comprising the administration of 40 sensory stimuli (encompassing 5 stimuli for every modality and intensity) while simultaneously recording EEG data. Participants were granted short breaks between blocks. Notably, the sequence of stimulus modality and intensity within each block was pseudo-randomized. Each trial started with a fixation cross displayed for 3 s and followed by a transient sensory stimulus. After a 3 s interval, participants were asked to verbally rate the perceived intensity on a 0–10 NRS within 5 s. A subsequent trial would initiate after 1–3 s following the rating period, resulting in an interstimulus interval ranging from 12 to 14 s.

In Dataset 4, participants received 40 stimuli, each with one of four stimulus intensities (i.e., 2.5, 3.0, 3.5, and 4.0 J), consisting of 10 laser pulses per intensity. The sequence of stimulus intensity was pseudo-randomized as well. The interstimulus interval varied randomly within 10–15 s. An auditory tone was presented 3–6 s after the stimulus, prompting participants to rate the intensity of the pain sensation elicited by the laser stimulus using the 0–10 NRS. In Dataset 5, participants received 80 brief laser stimuli at 4 different intensities (i.e., 2.5, 3.0, 3.5, and 4.0 J), for each of which a verbal pain rating (0–100) was collected.

A similar experiment with multiple levels of stimulus intensity was performed on patients in Dataset 6. For each patient, 10 laser pulses at one of six stimulus intensities (i.e., 3.0, 3.25, 3.5, 3.75, 4.0, and 4.25 J) were delivered on each side (i.e., pain-affected or unaffected side). The order of stimulus energies was pseudo-randomized, and the interstimulus interval varied randomly between 10 and 15 s. The participants would rate the intensity of pain sensation following each laser stimulus using the 0–10 NRS.

### Sensory stimulation

In Datasets 1–3, participants received transient stimuli of four sensory modalities: nociceptive laser, nonnociceptive tactile, auditory, or visual. Each sensory modality encompassed two levels of stimulus intensity (i.e., high and low).

Following each stimulus, participants were asked to verbally rate the perceived intensity on NRS from 0 ("no sensation") to 10 ("the strongest sensation imaginable"). The intensities of tactile, auditory, and visual stimuli were carefully set based on a pilot experiment, aiming to elicit perceived ratings of ~4 and 7 out of 10 for low and high intensities, respectively. However, due to the potentially unbearable nature of the 4.0 J nociceptive laser stimulus for certain participants, a categorizing approach was adopted. Specifically, participants were categorized into high and low pain sensitivity groups. High pain sensitivity group comprised individuals who rated the 4.0 J laser stimulus as above 8 out of 10 in the pilot experiment (Dataset 3). Conversely, participants who rated the 4.0 J stimulus as 8 or below were assigned to the low pain sensitivity group (Datasets 1 and 2).

The painful stimuli were delivered as short pulses of radiant heat (wavelength: 1.34 μm; pulse duration: 4 ms), generated by an infrared neodymium yttrium aluminum perovskite (Nd:YAP) laser (Electronic Engineering, Italy). The laser beam, transmitted through a 7 mm diameter optic fiber, targeted a pre-defined $5 \times 5\ cm^2$ area. To mitigate the risk of nociceptor fatigue or sensitization, the placement of laser beam was randomly adjusted by about 1 cm after each pulse. For Datasets 1 and 2, which comprised participants with low pain sensitivity, stimulus energies of 3.5 and 4.0 J were used. For Dataset 3, which involved participants with high pain sensitivity, laser stimuli were delivered with stimulus energies of 3.0 and 3.5 J. For Datasets 4 and 5, participants received nociceptive laser stimuli at four intensity levels: 2.5, 3.0, 3.5, and 4.0 J. For Dataset 6, six intensity levels of laser stimuli were delivered: 3.0, 3.25, 3.5, 3.75, 4.0, and 4.25 J.

Nonnociceptive tactile stimuli consisted of constant current square-wave electrical pulses (with a duration of 1 ms from model DS7A, Digitimer, UK), delivered through skin electrodes positioned 1 cm apart on the left wrist, over the superficial radial nerve. All three datasets used the same two stimulus intensities: 2.0 and 4.0 mA. Auditory stimuli consisted of brief pure tones (with 800 Hz central frequency, 50 ms duration, and 5 ms rise and fall times) delivered through headphones. All participants were exposed to the same two intensities (76 or 88 dB sound pressure level, SPL). Visual stimuli took the form of brief flashes of a gray round disk presented on a black background (with a duration of 100 ms) on a computer screen. The intensities were adjusted using the grayscale of the disk, with RGB values set at (100, 100, 100) and (200, 200, 200).

### EEG recording and preprocessing

EEG data were acquired via 64 Ag/AgCl electrodes positioned according to the International 10–20 System, using the nose as the reference (band-pass filter: 0.01–100 Hz; sampling rate: 1,000 Hz; BioSemi EEG system, the Netherlands for Dataset 1; Brain Products EEG system, Germany for Datasets 2–6). Electrode impedance was kept below 10 kΩ.

Electrooculographic signals were simultaneously recorded using two surface electrodes, one placed ~10 mm below the left eye and the other placed ~10 mm from the outer canthus of the left eye.

EEG data were preprocessed in MATLAB (R2021b; MathWorks, USA) using the EEGLAB toolbox v2023.1 [82]. Two preprocessing pipelines were adopted. For Datasets 1, 2, 3, 4, and 6, continuous EEG data were band-pass filtered between 1 and 30 Hz. Subsequently, EEG data were segmented into epochs extending from 500 ms before to 1,000 ms after stimulus onset. Each epoch was baseline corrected using the pre-stimulus interval. Trials contaminated by eye blinks and movements were corrected using an independent component analysis (ICA) algorithm implemented in the EEGLAB toolbox [82]. For Dataset 5, we applied a fully automated EEG pipeline DISCOVER-EEG, which was developed by the team who publicly shared Dataset 5 [83]. This pipeline included eight steps: (1) line noise removal with frequency-domain (multi-taper) regression as implemented in the EEGLAB function *pop_cleanline()*; (2) band-pass filtering between 1 and 100 Hz; (3) bad channel rejection with the EEGLAB function *pop_clean_rawdata()*; (4) re-referencing to average; (5) removal of independent components with ≥80% probability of being eye movement and muscle artifacts as identified by the ICLabel classifier [84]; (6) bad channel interpolation with spherical splines; (7) bad time segments removal with the Artifact Subspace Reconstruction method [85]; and (8) epoching from −500 ms to 1,000 ms. Default parameters were used for line noise removal, bad channel rejection, ICA, and bad time segments removal. The epoched data were then low-pass filtered at 30 Hz to remove high-frequency noise. Although the preprocessing pipeline applied to Datasets 1, 2, 3, 4, and 6 aligned with other EEG research [86,87], the more aggressive pipeline for Dataset 5 ensured that our findings would be robust across various preprocessing pipelines. Note that 15 participants were removed in Dataset 5 because they had <60 full trials (i.e., <15 trials per intensity condition on average) after bad time segments removal.

## Neural variability

We mainly used temporal SD, a variance-based neural variability measure at the single-trial level and then averaged across trials of the same stimulus intensity. Temporal SD is defined as the standard deviation of EEG responses within a time window:

$$tempSD = \sqrt{\frac{\sum_{i=1}^{n} (x_i - \bar{x})^2}{n}}$$

where $x_i$ is the amplitude at time point $i$, $\bar{x}$ is the mean amplitude within the time window of $n$ time points. We applied the sliding window of 100 ms ($n = 100$) to delineate how temporal SD changes over time (as shown in Fig 1C). Additionally, to evaluate the impact of time window duration on the results, we repeated analyses using time windows of 50 and 200 ms. Please note that the coefficient of variation (CV), which normalizes the SD by dividing it by the mean, is another common measure of neural variability [8]. However, CV becomes unstable when the mean is small, as empirically demonstrated in Dataset 1 (S11 Fig). Therefore, we did not employ this metric in the present study.

We also used an information theory-based measure, PE, as an additional metric of neural variability to demonstrate the robustness of the relationship between neural variability and pain intensity discriminability. PE quantifies the complexity of a time series by evaluating the order relations between values in the time series rather than their absolute numerical values [42]. We computed PE with a customized function *PE* from the MATLAB file exchange [88]. The computation process involves selecting an embedding dimension $m$, representing the number of sequential time points forming a pattern, and a time delay $\tau$. Then vectors $v_t = (x_t, x_{t+\tau}, x_{t+2\tau}, \ldots, x_{t+(m-1)\tau})$ are extracted from a windowed time series $(x_1, x_2, \ldots, x_n)$, where $n$ denotes the number of time points within the specified window. These vectors facilitate the enumeration of relative frequencies for all potential ordinal patterns by ranking the elements to construct a probability distribution $P$ across all patterns. Then PE can be calculated using Shannon entropy as follows:

$$PE = -\sum_{p \in P} p \cdot log(p)$$

In our case, we set $m = 4$, $\tau = 1$, and $n = 100$. Therefore, 97 vectors can be extracted from a time window of 100 time points, namely, $v_1 = (x_1, x_2, x_3, x_4)$, $v_2 = (x_2, x_3, x_4, x_5)$, ..., $v_{97} = (x_{97}, x_{98}, x_{99}, x_{100})$. We then determined the relative frequencies of all possible ordinal patterns within these vectors, such as, (1, 2, 3, 4), (1, 2, 4, 3), (1, 3, 2, 4), etc. and calculated Shannon entropy to obtain PE for each time window. Similar to the temporal SD calculation, we also employed the sliding window technique to characterize the temporal changes in PE.

### Sensory discriminability

Sensory discriminability was quantified by the difference between ratings averaged across trials of high intensity and trials of low intensity. A previous study has shown that this simple index has a less skewed distribution and yields results highly similar to those of other indices, such as measures based on signal detection theory (SDT) [21]. Different individuals, however, often use rating scales differently, which constitutes another source of interindividual variability. To control for this internal scaling bias, we conducted a control analysis utilizing a discriminability index derived from SDT, specifically the AUC.

SDT decomposes sensory judgment into two orthogonal components, namely, scaling bias (e.g., the tendency to report higher ratings) and discriminability [89]. SDT-derived discriminability metrics such as AUC are thus less affected by the scaling bias and have been used in previous studies on discriminability [90,91]. Our primary discriminability measure, rating differences, is conceptually related to SDT-derived measures. Taking the rating difference between two stimuli can partially control for the internal scaling in some sense, so long as one makes the assumption that the internal rating scale remains similar across different stimulus intensities for the same individual. Formally, assume that the rating $r$ of a stimulus $s$ is the sum of the true perception $p$ and a constant rating bias $b$, namely, $r_s = p_s + b$, then the rating bias cancels out in the rating differences, that is, $r_{s1} - r_{s2} = (p_{s1} + b) - (p_{s2} + b) = p_{s1} - p_{s2}$. However, raw rating differences do not fully account for the effect of rating distributions such as variance and therefore are still biased. SDT-based metrics can better control the distribution differences by normalizing rating differences using distributional parameters (e.g., variance in the case of $d' = \frac{r_{s1} - r_{s2}}{\sqrt{0.5 \times (Variance_{s1} + Variance_{s2})}}$ and nonparametric control over distributions as in AUC). Our previous work has demonstrated that the choice of metrics (rating differences, $d'$, or AUC) did not affect the relationship between pain discriminability and EEG responses [21]. In the present study, we also found that the SDT-derived AUC measure correlated strongly with pain rating differences ($r = 0.832$), confirming that rating differences can control for, at least partly, the internal scaling effect.

### Statistical analysis

**Correlation analysis.** Correlation analyses were performed to test whether (1) neural variability reflects pain intensity discriminability, (2) the neural variability-intensity discriminability association is stimulus-general, (3) the relationship between neural variability and pain intensity discriminability is generalizable, and (4) neural variability encodes with interindividual sensory sensitivity and intraindividual sensory perception.

To determine if neural variability reflects pain intensity discriminability, we computed neural variability (i.e., temporal SD) at the single-trial level, took the difference of neural variability between two intensities (high − low, ΔSD), and correlated this neural variability difference with pain intensity discriminability in a point-by-point manner in Datasets 1–3. Multiple comparisons were corrected using the FDR procedure [92]. Scalp topographies were computed by spline interpolation. To illustrate the correlations in scatterplots, we extracted the values within a 20 ms window centered at the peak of the subject-averaged neural variability difference and correlated them with pain intensity discriminability. Note that we applied FDR corrections across time points for electrode Cz only. Topographical maps are presented for visualization, and no spatial statistical tests were conducted, as our conclusions do not depend on spatial distributions.

To test whether the relationship between neural variability and sensory discriminability is stimulus-general, we repeated correlation analyses and partial correlations controlling for ERP amplitude on Dataset 1–3 with sensory stimuli of tactile, auditory, and visual modalities. To directly compare pain and nonpain modalities, partial correlational coefficient differences were tested with Steiger's z test [93]. To control for rating differences between modalities, we also manually matched ratings between pain and nonpain modalities (see Rating matching approach below), and conducted correlation analyses on the matched data.

To demonstrate the generalizability of our findings, we further explored the correlations between neural variability differences and pain intensity discriminability in Datasets 4–6, all of which were recorded from participants exposed to various intensities of nociceptive stimuli. In Datasets 4 and 5, four levels of stimuli with a 0.5 J interval were used, resulting in six pairs of high- and low-intensity combinations. We first applied one-way repeated measures ANOVA to determine whether ratings of pain intensity differed across stimulus intensities. We then performed paired-sample t tests on temporal SD evoked by various pairs of stimulus intensities. For each pair of stimulus intensities, we performed correlation analyses as previously described between neural variability and pain intensity discriminability. In Dataset 6, six levels of stimuli separated by a 0.25 J interval were applied on both the affected and unaffected sides, resulting in 15 pairs of high- and low-intensity combinations for each side. We first applied two-way repeated measures ANOVA to determine whether ratings of pain intensity were significantly different across stimulus intensities and stimulation sides. We then performed paired-sample t tests on ratings and temporal SD values of different pairs of stimulus intensities. Spearman's rank correlation analyses were then conducted on those pairs of high- and low-intensity combinations that showed significant differences in both ratings and temporal SD.

In addition to intensity discriminability, we also tested how interindividual sensory sensitivity and intraindividual sensory perception relate to neural variability in Datasets 1–3. Interindividual sensory sensitivity was operationalized as mean intensity ratings across the high and low intensity conditions ([High+Low]/2) for each subject. Mean ratings were then correlated with mean temporal SD across the high and low intensity conditions. Intraindividual sensory perception was quantified as single-trial intensity ratings, which were correlated with single-trial temporal SD for each subject. The subject-wise correlation coefficients were transformed to z values using the Fisher z-transformation (z-Fisher), and the z values were finally compared against zero using a one-sample t test.

To provide direct evidence for or against null hypotheses, we also computed BF for the foregoing tests under Jeffreys-Zellner-Siow priors with the bayesFactor toolbox (ver 1.0.0; https://github.com/klabhub/bayesFactor) [94]. BF quantifies the relative likelihood of collecting the data given the alternative hypothesis over the null hypothesis [95]. Different BF values were interpreted in the way recommended by Kass and Raftery [96]: (1) 1–3.3: barely worth mentioning evidence for the alternative hypothesis; (2) 3.3–10: substantial evidence; (3) 10–100: strong evidence; and (4) >100: decisive evidence. The reciprocal of BF values less than 1 can be interpreted as evidence for the null hypothesis. For example, a BF of 0.2 (i.e., 1/5) indicates substantial evidence for the null hypothesis.

Note that all statistical tests were two-tailed.

**Rating matching analysis.** Perceptual ratings can vary significantly between different sensory modalities, making it challenging to compare the relationship between EEG responses and discriminability indices across modalities. This variability in perceptual ratings might confound any observed differences between modalities. To account for this potential confounder, we used a rating matching procedure to equalize intensity ratings between pairs of sensory modalities (i.e., pain versus touch, pain versus audition, and pain versus vision) in both Dataset 1 and Datasets 2&3.

The matching procedure was adapted from a previous study [68]. When matching pain and touch ratings, if Participant-X rates high- and low-intensity laser stimuli as, for instance, 6 and 4 on average, respectively, we identify participants whose mean ratings for high-intensity tactile stimuli fall within 5.5–6.5 and for low-intensity tactile stimuli within 3.5–4.5. This ensures an absolute average matching error of ≤0.5 for both high- and low-intensity stimuli. If M participants meet this criterion, we further narrow it down to N participants with the smallest absolute matching error. If N equals

1, Participant-X is paired with this single participant. If $N$ is greater than 1, Participant-X is randomly paired with one of these participants (say, Participant-Y), provided that Participant-Y has not been paired with anyone else. The effect of this matching procedure was confirmed by statistically insignificant interaction effects (sensory modality × stimulus intensity) in mixed-design ANOVA.

**Independence test.** Since variance and mean amplitude of the brain response are generally not independent, we first correlated neural variability differences and amplitude differences between high- and low-intensity conditions (also referred to as ΔSD and ΔAmplitude, respectively), and then performed partial correlation analysis between ΔSD and pain intensity discriminability, controlling for ΔAmplitude at the same time point. Note that, consistent with neural variability, ERP amplitude was also calculated with a 100-ms sliding window. To further account for ΔAmplitude, we subtracted trial-averaged amplitude from single-trial EEG responses for each condition to derive induced EEG responses [43], and then computed neural variability and conducted correlation analysis with the induced EEG responses. To illustrate the correlations in scatterplots, we also extracted the values within a 20-ms window centered at the peak of the subject-averaged neural variability difference. After regressing out the ΔAmplitude, scatterplots were created using the residuals of ΔSD and pain intensity discriminability.

**Noise simulation analysis.** To examine the effect of differential noise levels in low- and high-intensity conditions, we conducted simulation analyses. Noise was sampled from baseline (−500 to 0 ms) in single-trial EEG signals in the low- and/or high-intensity conditions. This approach ensures that the noise structure in real EEG signals is respected and that residual noise after preprocessing such as muscle artifacts, baseline drift, and amplifier noise can be captured. The sampled noise was then scaled by a factor (referred to as "noise scale") and added to post-stimulus EEG signals. We conducted three sets of simulation: (1) adding noise to the low-intensity condition alone, (2) high-intensity condition alone, and (3) both low- and high-intensity conditions. Eleven noise scales were chosen: 0–2 in steps of 0.2. Despite appearing small, a noise scale of 2 completely changed neural variability as measured by SD (S8A and S8C Fig). Since this level of noise seemed empirically unlikely, our analyses could also serve as a stress test to demonstrate the robustness of our findings.

**Dominance analysis.** To assess the contribution of differential neural variability and differential mean amplitude (i.e., ΔSD and ΔAmplitude) to encoding pain intensity discriminability, we conducted dominance analysis at each time point [44,45]. Dominance analysis is a method for comparing the relative contribution of predictors in multiple regression models. We quantified contribution using total dominance, which has the desirable property that the sum of total dominance of all predictors is equal to the determination coefficient ($R^2$) of the full multiple regression model. At each time point, we first regressed pain intensity discriminability against differential neural variability and differential mean amplitude, and computed $R^2$ of this full model, which is denoted by $R^2_{full}$. Then, we predicted pain intensity discriminability separately using differential neural variability and differential mean amplitude. $R^2$ of these two reduced models is denoted by $R^2_{var}$ and $R^2_{amp}$. According to the dominance analysis approach, total dominance of neural variability differences is defined as $(R^2_{full} - R^2_{amp} + R^2_{var})/2$. Similarly, total dominance of mean amplitude differences is $(R^2_{full} - R^2_{var} + R^2_{amp})/2$.

**Resampling analysis.** We used a resampling method to examine the influence of subject number and trial number on the probability of detecting significant correlations between neural variability and pain intensity discriminability. In Datasets 1–3, we generated bootstrapped subsamples from the whole dataset, that is, randomly selected data samples with replacement. For the subject number, we subsampled subjects ranging from 20 to the maximum of each dataset, and repeated the resampling procedure for 100 times for each subject number. To assess the effect of trial number, we subsampled trials ranging from 1 to 15 (in steps of 1), and for each trial number, the resampling procedure was repeated for 100 times. Note that during the resampling of subjects, we used data from all trials, while during the resampling of trials, data from all participants were used. In each subsample, we computed differential neural variability and differential ERP amplitude, correlated them with pain intensity discriminability, and applied a threshold to the correlation time series with $p$(FDR) = 0.05. Since we generated 100 subsamples for each sample size, the probability of a time point

being significant under a certain subject/trial number was approximated by the number of times that time point reached significance. We also specifically examined the number of subjects/trials needed to detect significant correlations with a probability ≥80% [97].

**Oscillatory profiling analysis.** To further explore whether neural variability and ERP amplitude were driven by frequency-specified activity, we band-pass filtered single-trial EEG signals to characterize the oscillatory profiles of neural variability and ERP amplitude as indicators of pain intensity discriminability. Since continuous EEG traces were filtered from 1 to 30 Hz to derive ERPs, here we focused on four classical frequency bands: delta (1–4 Hz), theta (4–8 Hz), alpha (8–12 Hz), and beta (12–30 Hz). After band-pass filtering the data for the respective frequency band, we then repeated the partial correlational analyses, that is, correlating ΔSD in the filtered data with pain intensity discriminability in a point-by-point manner while controlling for ΔAmplitude at the same time point, and correlating ΔAmplitude in the filtered data with pain intensity discriminability while controlling for ΔSD at the same time point.

## Supporting information

**S1 Fig. Rating differences as slopes of psychophysical functions.** Given two fixed and medium physical intensities $x1$ and $x2$, the rating differences $y3 - y1$ and $y2 - y1$ can be viewed as the slope of psychophysical functions. Individuals with higher discriminability have a larger slope (i.e., $y3 - y1 > y2 - y1$).
(TIF)

**S2 Fig. Correlations between ERP amplitudes and pain intensity discriminability in Dataset 1. (A)** Comparison of ERP amplitudes derived from 100 ms sliding windows between conditions of high- and low-intensity stimuli in Dataset 1. **(B)** Correlation between amplitude difference and SD difference. Black segments of the curve illustrate time points where correlations were not significant after FDR correction, while red segments illustrate time points with significant correlations. **(C, D)** Correlations between ERP amplitude difference and pain intensity discriminability. ERP amplitude differences around N2 and P2 peaks were significantly correlated with pain intensity discriminability. **(E, F)** Partial correlations between ERP amplitude difference and pain intensity discriminability controlling for SD. The light-colored "bursty" curve in panel E represents the subject-averaged residuals of Δamplitude after regressing out ΔSD. Partial correlations were calculated while controlling for Δamplitude. Note that the gray bars represent Pearson's $r$ values at time points where significant correlations were observed after FDR correction. The color bars underneath display the corresponding BF values for the correlations. Part-corr is short for partial correlation. BF is short for Bayes factor. Error bars are 95% confidence intervals. The data underlying this Figure can be found in https://doi.org/10.17605/OSF.IO/QTV8A.
(TIF)

**S3 Fig. Information theory-based neural variability encodes pain intensity discriminability in Dataset 1. (A)** Neural variability assessed by an information theory-based measure, temporal PE of EEG responses evoked by high-intensity (violet) and low-intensity (gold) stimuli in Datasets 1. **(B)** Point-by-point correlations between neural variability differences (high−low, ΔPE) and rating differences (high−low, ΔRating). Note that the gray bars represent Pearson's $r$ values at time points where significant correlations were observed after FDR correction. The color bars underneath display the corresponding Bayes factor values for the correlations. **(C)** Correlations between values around the trough of temporal PE and ΔRating. Error bars are 95% confidence intervals. The data underlying this Figure can be found in https://doi.org/10.17605/OSF.IO/QTV8A.
(TIF)

**S4 Fig. Correlations between neural variability of induced responses and pain intensity discriminability in Dataset 1. (A)** Temporal SD of induced EEG responses and correlations between ΔSD and pain intensity discriminability in Dataset 1. **(B)** Temporal PE of induced EEG responses and correlations between ΔPE and pain intensity discriminability

in Dataset 1. Note that the gray bars represent Pearson's *r* values at time points where significant correlations were observed after FDR correction. The color bars underneath display the corresponding Bayes factor values for the correlations. Error bars are 95% confidence intervals. The data underlying this Figure can be found in https://doi.org/10.17605/OSF.IO/QTV8A.
(TIF)

**S5 Fig. Correlations between neural variability and pain intensity discriminability measured by AUC in Dataset 1. (A)** Neural variability measured by temporal SD and its partial correlation with pain intensity discriminability measured by AUC controlling for the amplitude of ERPs. **(B)** Neural variability measured by temporal PE and its partial correlation with AUC controlling for the amplitude of ERPs. Note that the gray bars represent Pearson's *r* values at time points where significant correlations were observed after FDR correction. The color bars underneath display the corresponding Bayes factor values for the correlations. Part-corr is short for partial correlation. The light-colored "bursty" curves represent the subject-averaged residuals of SD/PE after regressing out ERP amplitude. The significance of both indices demonstrates the stability and robustness of neural variability as an indicator of pain intensity discriminability. In scatterplots, *r* represents *r* value of partial correlation and error bars are 95% confidence intervals. The data underlying this Figure can be found in https://doi.org/10.17605/OSF.IO/QTV8A.
(TIF)

**S6 Fig. Correlations between pain intensity discriminability and neural variability calculated with different lengths of sliding windows in Dataset 1. (A, B)** Partial correlations of neural variability and pain intensity discriminability controlling for ERP amplitude. The neural variability was calculated with sliding windows of 50 ms in (A) and 200 ms in (B). Consistent significant findings existed in both window sizes. Note that the gray bars represent Pearson's *r* values at time points where significant correlations were observed after FDR correction. The color bars underneath display the corresponding Bayes factor values for the correlations. The light-colored "bursty" curves represent the subject-averaged residuals of SD/PE after regressing out ERP amplitude. In scatterplots, *r* represents *r* value of partial correlation and error bars are 95% confidence intervals. The data underlying this Figure can be found in https://doi.org/10.17605/OSF.IO/QTV8A.
(TIF)

**S7 Fig. Comparable results from single-trial-based and average trial-based analyses to calculate neural variability for pain in Dataset 1. (A, B)** Neural variability based on single-trial SD (A) and average trial SD (B) showed similar partial correlations with pain intensity discriminability after regressing out mean ERP amplitude. Note that the gray bars represent Pearson's *r* values at time points where significant correlations were observed after FDR correction. Part-corr is short for partial correlation. The data underlying this Figure can be found in https://doi.org/10.17605/OSF.IO/QTV8A.
(TIF)

**S8 Fig. Effects of adding different levels of noise to EEG data on correlations between pain intensity discriminability and neural variability in Dataset 1. (A)** ΔSD time series when different levels of noise were added to the low-intensity condition. As the added noise level increased, SD in the high-intensity condition became smaller than that in the low-intensity condition. Noise scale represents the level of noise added. Eleven noise scales (0–2 in steps of 0.2) were tested. A noise scale of *s* means *s* times the baseline noise in single-trial EEG signals was added. **(B)** Bayes factor (BF) for partial correlations between pain intensity discriminability and neural variability when noise was added to the low-intensity condition while controlling for ERP amplitude differences. Enclosed areas represent correlations with $\log_{10}(BF)$ ≥ 0.5, namely, BF ≥ 3.3. Adding noise to the low-intensity condition had no substantial effect on the correlation between pain intensity discriminability and neural variability. **(C)** ΔSD time series when different levels of noise were added to the high-intensity condition. **(D)** BF for partial correlations between pain intensity discriminability and neural variability when different levels of noise were added to the high-intensity condition while controlling for ERP amplitude differences. Adding

noise to the high-intensity condition also had no substantial effect on the correlation between pain intensity discriminability and neural variability. (**E**) ΔSD time series when different levels of noise were added to the high- and low-intensity conditions simultaneously. (**F**) BF for partial correlations between pain intensity discriminability and neural variability when different levels of noise were added to the high- and low-intensity conditions while controlling for ERP amplitude differences. Adding noise also had no substantial effect on the correlation between pain intensity discriminability and neural variability. The data underlying this Figure can be found in https://doi.org/10.17605/OSF.IO/QTV8A.
(TIF)

**S9 Fig. Comparisons of neural variability and ERP amplitude at different frequency bands for encoding pain intensity discriminability in Datasets 2 and 3. (A–D)** Differential amplitude (black curves), neural variability (blue curves) at the delta (1–4 Hz), theta (4–8 Hz), alpha (8–12 Hz), and beta (12–30 Hz) bands, and their partial correlations with pain intensity discriminability while controlling for each other in Datasets 2 and 3. Note that the gray bars represent Pearson's *r* values at time points where significant correlations were observed after FDR correction. The color bars underneath display the corresponding Bayes factor values for the correlations. The topographies represent partial r values within a ±10 ms window around the peak. For ΔAmplitude calculations in A and B, the latencies of N2 and P2 peaks were utilized, while for C and D, where clear ERPs were not observed, latencies corresponding to the minimal and maximal values within the 100–500 ms time window were used. The data underlying this Figure can be found in https://doi.org/10.17605/OSF.IO/QTV8A.
(TIF)

**S10 Fig. Point-by-point partial correlations between neural variability of EEG responses evoked by tactile, auditory, and visual stimuli and their respective sensory intensity discriminability. (A–C)** Neural variability, and its partial correlations with sensory intensity discriminability of tactile, auditory, and visual stimuli in Dataset 1. Gray bars represent Pearson's *r* values at time points where significant correlations were observed (FDR-corrected). The color bars underneath display the corresponding Bayes factor values for the correlations. The light-colored "bursty" curves represent the subject-averaged residuals of ΔSD after regressing out Δamplitude at each time point. For tactile modality, significant partial correlations while controlling for mean amplitude differences were found at 35–180 ms, but no significance was observed around the peak of neural variability (see Fig 5). No significance was observed for either auditory or visual modalities in partial correlations while controlling for mean amplitude differences of ERPs. **(D–F)** Neural variabilities and their partial correlations with respective sensory intensity discriminability in Datasets 2&3. No significant partial correlations while controlling for mean amplitude differences were observed for any of these sensory modalities. Part-corr is short for partial correlation. The data underlying this Figure can be found in https://doi.org/10.17605/OSF.IO/QTV8A.
(TIF)

**S11 Fig. Instability of coefficients of variation as a measure of neural variability of pain in Dataset 1.** The coefficient of variation (CV) normalizes SD by dividing it by the mean. CV was extremely unstable in many time points. Note that the *y* axis is capped at ±150 for better visualization. Actually, CV values ranged from a maximum of 2,442 to a minimum of −5,017, illustrating its instability in conditions with low mean signal amplitudes. The data underlying this Figure can be found in https://doi.org/10.17605/OSF.IO/QTV8A.
(TIF)

## Author contributions

**Conceptualization:** Li-Bo Zhang, Xin-Yi Geng., Li Hu.

**Data curation:** Li-Bo Zhang.

**Formal analysis:** Li-Bo Zhang, Xin-Yi Geng.

**Funding acquisition:** Li Hu.

**Investigation:** Li-Bo Zhang, Xin-Yi Geng.

**Methodology:** Li-Bo Zhang, Xin-Yi Geng, Li Hu.

**Project administration:** Li-Bo Zhang, Li Hu.

**Software:** Li-Bo Zhang, Xin-Yi Geng.

**Supervision:** Li Hu.

**Validation:** Li-Bo Zhang, Xin-Yi Geng.

**Visualization:** Li-Bo Zhang, Xin-Yi Geng, Li Hu.

**Writing – original draft:** Li-Bo Zhang, Xin-Yi Geng.

**Writing – review & editing:** Li-Bo Zhang, Xin-Yi Geng, Li Hu.

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
