## [Editor Report · Decision Letter 0]

21 Aug 2025

Dear Dr Hu,

Thank you for submitting your revised manuscript entitled "Neural variability reliably and preferentially encodes pain intensity discriminability" for consideration as a Research Article by PLOS Biology.

Your manuscript has now been evaluated by the PLOS Biology editorial staff and I am writing to let you know that we would like to send your submission back for external peer review.

Once your full submission is complete, your paper will undergo a series of checks in preparation for peer review. After your manuscript has passed the checks it will be sent out for review. To provide the metadata for your submission, please Login to Editorial Manager (https://www.editorialmanager.com/pbiology) within two working days, i.e. by Aug 23 2025 11:59PM.

Kind regards,

Christian

Christian Schnell, Ph.D.

Senior Editor

PLOS Biology

cschnell@plos.org

on behalf of

Taylor Hart, PhD,

Associate Editor

PLOS Biology

thart@plos.org

---

## [Decision Letter · Decision Letter 1]

3 Oct 2025

Dear Dr Hu,

Thank you for your patience while we considered your revised manuscript "Neural variability reliably and preferentially encodes pain intensity discriminability" for publication as a Research Article at PLOS Biology. This revised version of your manuscript has been evaluated by the PLOS Biology editors, the Academic Editor and several independent reviewers.

Based on the reviews, we are likely to accept this manuscript for publication, provided you satisfactorily address the remaining points raised by the reviewers. Please also make sure to address the following data and other policy-related requests.

IMPORTANT: Your next revision should address the following editorial requests so as to avoid delay in publication:

----------

**Title

Based on Reviewer 2's feedback and ease of accessibility, we would like to tweak your paper's title. We suggest the following alternative title:

"Neural variability reliably encodes interindividual differences in the perception of pain intensity"

**Abstract

As requested by Reviewer 2, please remove Point 3 from the Abstract.

**Ethics:

Please create a new sub-heading for the Materials and Methods section titled "Ethics Statement". This sub-heading should contain all of the information about participant consent, ethical guidelines, and approvals.

Please also include the full name of all relevant Institutional Review Boards (or equivalent) and approval numbers. Please note that these experiments and must have been conducted according to the principles expressed in the Declaration of Helsinki.

**Data and Code availability:

-- Thank you for uploading your data and code. Please make them accessible so that we can examine them.

-- Please also supply the numerical values necessary to reproduce the main findings, either in a supplementary excel file or as a permanent DOI’d deposition for the following figures:

6ABCDE

7AC

8A

-- Please cite the location of the data clearly in all relevant main and supplementary Figure legends, e.g. “The data underlying this Figure can be found in S1 Data” or “The data underlying this Figure can be found in https://doi.org/10.5281/zenodo.XXXXX”

----------

We expect to receive your revised manuscript within two weeks.

*Published Peer Review History*

*Press*

Sincerely,

Taylor

Taylor Hart, PhD,

Associate Editor

thart@plos.org

PLOS Biology

Reviewer #1: General Comments:

The authors have made commendable efforts to address the concerns raised in the initial review. They have performed numerous additional control analyses, which have considerably strengthened the manuscript. The replies to my previous comments are thoughtful and mostly convincing, and overall, the manuscript is now much improved in clarity and rigor.

A few issues remain where I believe further clarification or moderation would improve the paper. These relate to the interpretation of modality specificity, the strength and clinical relevance of the patient findings, the definition and interpretation of discriminability, and the scope of what the results mean.

Specific Comments:

1. Specificity of pain-related discriminability

The authors argue for a modality-specific role of neural variability in pain discriminability. While the new analyses are convincing, I still find it conceptually difficult to believe that fundamentally different mechanisms underlie discriminability for pain compared to other sensory modalities. I do not see a straightforward way to further test this empirically with the present dataset, but it may help if the discussion more fully considers this phenomenon, including alternative interpretations.

2. Findings in patients and clinical relevance

The analyses in patients are an important addition, but the relatively small sample size (N = 27) limits the strength of these conclusions. Moreover, no direct statistical comparison has been performed between patients and healthy participants. I therefore encourage the authors to present these results more cautiously, particularly in the abstract and conclusion. It would also be helpful to explicitly report the number of patients in the abstract. Similarly, the potential clinical relevance could be moderated to reflect the preliminary nature of the patient data.

3. Definition and operationalization of discriminability

I remain somewhat puzzled by the definition of discriminability as differences in subjective rating scales. While I do not see a specific error in the reasoning, it remains unclear whether this measure truly captures the ability to discriminate as opposed to differences in internal scaling or rating behavior. The additional analyses based on signal detection theory (SDT) are a valuable step forward, though I am not in a position to fully evaluate their validity. I encourage the authors to explain the rationale for these analyses and their implications more clearly so that the broader readership can follow the argumentation. In addition, it would be helpful to explain the authors' definition of discriminability and its behavioral relevance more explicitly. For example, what does it mean in everyday life when one individual rates two stimuli as 3 and 5, while another individual rates the same stimuli as 3 and 7? Finally, the abstract should make it clear that the findings relate to differences in discriminability between participants (i.e., inter-individual differences) rather than discriminability between trials.

Reviewer #2: The research 'Neural variability reliably and preferentially encodes pain intensity discriminability' comprises a deep and considered analysis of measures of neural variation in relation to pain discriminability (as quantified by objective differences in stimulation levels) across multiple datasets. The manuscript is well written and the rationale for the research is clear. The authors have made considered responses to previous reviews which, generally, explain their decisions well and satisfy concerns. I note a few minor points, particularly regarding the strength of interpretation in key aspects in cluding title and abstract, which warrant further consideration before the manuscript is suitable for publication.

I largely agree with R1 (Point 2) and R2 (point 2) that the links made between chronic pain and the outcomes of this research are preliminary and, as such, they should be described and discussed carefully. The authors have made some effort reduce and improve this in their latest version and have included more up-to-date studies. However,some aspects still press this point that neural variability in response to experimental pain could offer immediate insights into chronic pain mechanisms or clinical utility. Specifically, I think point 3 in the abstract is overstatement and I recommend that this should be fully removed. The present research includes just one small PHN group and I would agree with the authors own interpretation employed later in the manuscript proper were they state that this offers only preliminary support towards links to chronic pain or clinical utility. It is a discussion point but not something worthy of claiming in abstract.

I also feel that there is still some overstatement of the findings in relation to pain selectivity. R3, myself (and I think many readers) will still have concerns about the difficulty of matching salience across modalities. I would argue that the authors finding that discriminability-variability links in pain modality could exhibit a preferential status can be described as preliminary or requiring further research. I believe readers will still hope to see future independent replication and investigations with specific experimental designs to directly consider stimuli salience. The author response to R3 about this same point 'in the interpretation of our findings, we have replaced the term "pain selectivity" with the more neutral "pain preferential"' is not really sufficient in my estimation. One direct change wI recommend is to to cut 'preferential' from the title. The link between neural variability and pain discriminability is enough for title and I do not believe that this work is yet sufficient to claim that this relationship is definitively preferential as the title suggests. Likewise, in the abstract where they state 'and preferential indicator' this should be more nuanced and changes to 'potentially preferential indicator' I much prefer the authors language in results: 'the results above offer preliminary evidence that neural variability preferentially encodes pain intensity discriminability.' This is how I think it could be pitched throughout the manuscript. On line 331 I suggest changing 'probably' to 'possibly' due to the need for further research with regards to salience and other factors. These minor changes, along with above changes to title and abstract, will better align the data and claims in the manuscript for the reader.

---

## [Editor Report · Decision Letter 2]

17 Oct 2025

Dear Dr Hu,

Thank you for the submission of your revised Research Article "Neural variability reliably encodes interindividual differences in the perception of pain intensity" for publication in PLOS Biology. On behalf of my colleagues and the Academic Editor, Choong-Wan Woo, I am pleased to say that we can in principle accept your manuscript for publication, provided you address any remaining formatting and reporting issues. These will be detailed in an email you should receive within 2-3 business days from our colleagues in the journal operations team; no action is required from you until then. Please note that we will not be able to formally accept your manuscript and schedule it for publication until you have completed any requested changes.

PRESS

Sincerely, 

Taylor

Taylor Hart, PhD,

Associate Editor

PLOS Biology

thart@plos.org